# HuRi: Humanoid Robots Adaptive Risk-aware Distributional Reinforcement Learning for Robust Control

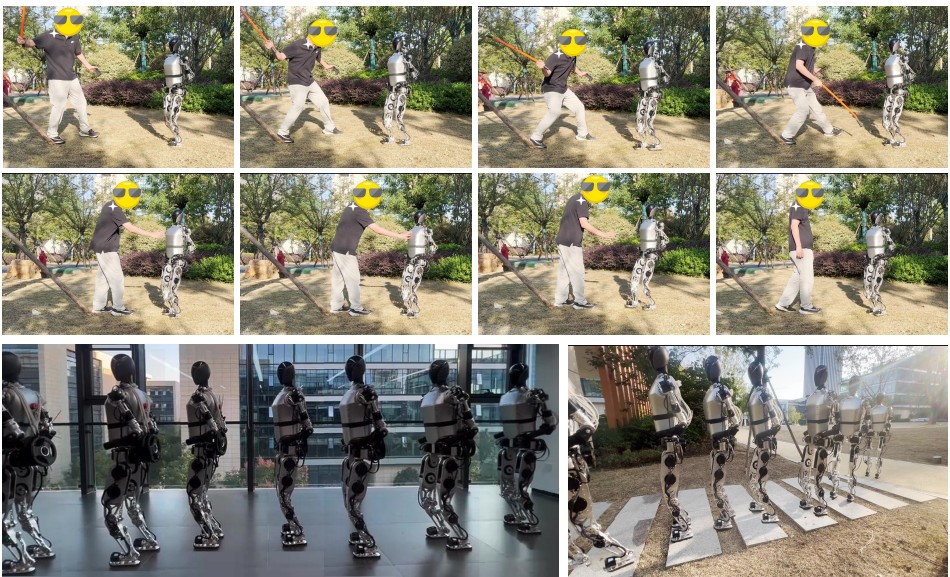

Figure 1: We use risk-aware distributional reinforcement learning algorithm(HuRi) to train a robust locomotion control policy that can be deployed on a physical robot —— Zerith-1.

## Abstract

Humanoids Locomotion remains an unsolved challenge, primarily due to the significantly smaller stability margin compared to other types of robots. As a result, the control systems for humanoid robots must place greater emphasis on risk mitigation and safety considerations. Existing studies have explicitly incorporated risk factors into robot policy training, but lacked the ability to adaptively adjust the risk sensitivity for different risky environment conditions. This deficiency impacts the agent's exploration during training and thus fail to select the optimal action in the risky environment. We propose an adaptive risk-aware policy(HuRi) based on distributional reinforcement learning. In Dist. RL, the policy control the risk sensitivity by employing different distortion measure of the esitimated return distribution. HuRi is capable of dynamically selecting the risk sensitivity level in varying environmental conditions by utilizing the Inter Quartile Range to measure intrinsic uncertainty and Random Network Distillation for assessing the parameter uncertainty of the environment. This framework allows the humanoid to model the uncertainty in the environment and then conduct safe and efficient exploration in hazardous environments; therefore enhancing the mobility and adaptability of humanoid robots. Simulations and real-world experiments on the Zerith-1 robot have demonstrated that our method could achieve significantly more robust performance, compared to other methods, including ablated versions.

# 1 INTRODUCTION

Humanoid robots, with their human-like appearance and potential for strong motor capabilities, have garnered extensive research interest. They are expected to operate in complex and hazardous environments, replacing humans in performing tasks. A fall or accident can result in task failure or even hardware damage. Particularly in risk-prone environments characterized by high uncertainty, the risk of accidents involving humanoid robots escalates significantly. Consequently, ensuring their safe operation becomes paramount.

Recent advancements in Deep Reinforcement Learning control have enabled legged robots to traverse difficult terrains Zhuang et al. (2023); Cheng et al. (2023). Although these methods strive to improve the locomotion capabilities of robots, they do not explicitly model environmental risks. Distributional reinforcement learning(Dist. RL) models the whole distribution of returns rather than merely their expected value. It learns a parameterized return distribution and optimizes the loss function, capturing more information about return uncertainty. This approach is especially valuable in scenarios where effective risk management is essential.

Many methods simulate the stochastic uncertainty in the environment by learning a probabilistic distribution through quantile regression and executing risk-averse policies by optimizing for worst-case scenarios based on risk distortion measures. However, in these methods, agents maintain a fixed risk sensitivity in dynamic environment, which may lead to suboptimal result. In addition, maintaining a constant level of risk sensitivity throughout the training process can cause the agent to exhibit excessively conservative behavior in some situations. This excessive caution can lead the agent to shy away from actions that appear risky, even if they could yield substantial long-term gains. As a result, having a fixed risk sensitivity can result in suboptimal exploration, with the agent becoming reliant on local optima. This approach can make the agent inflexible when confronted with varying environment conditions, thereby diminishing the model's overall adaptability and performance. The key focus of this research is to explore how to achieve safe exploration during training and to enhance the agent's ability to resist out-of-distribution disturbances in risky scenarios.

In this research, we propose the **HuRi** method, which explicitly evaluates the risks of humanoid robot locomotion using Dist. RL, without relying on external devices such as unreliable cameras. When the agents interact with the environment, Dist. RL models the return distribution, reflecting the inherent uncertainty in the system, which we can leverage to assess and optimize policies. In Dist. RL, the agent's risk sensitivity can be controlled by applying different distortion measures to the computed return distribution. Unlike previous robot locomotion control methods, we incorporate random network distillation to measure parameter uncertainty and interquartile range to quantify the environment's intrinsic uncertainty, adaptively adjusting the scalar risk parameter of the distortion function. This adaptive adjustment allows the robot to select different risk sensitivity levels in varying environment conditions. HuRi is capable of adaptively perceiving environmental uncertainty, advocating for more cautious behavior in states that are seldom visited and encouraging the exploration of more promising actions in familiar. This capability is instrumental in enabling agents to accommodate various environmental changes, deeply explore dynamic risk environments, and resist out-of-distribution disturbances.

To the best of our knowledge, we are the first to propose an adaptive risk-aware policy learning method in the field of humanoid robots. Through both simulation and real-world experiments, we verified the effectiveness of our method in risky scenarios compared with other methods. Our approach significantly improves the robustness of humanoid robot locomotion. Our primary contributions are as follows:

- We innovatively propose an adaptive risk-aware distributional reinforcement learning policy that enables agents to adjust the risk preference of the policy, thereby promoting safe and efficient exploration during training and enhancing the agent's performance.

- We explicitly model risk factors in humanoid robot locomotion control, enabling agents to resist environmental stochastic disturbances in dynamic risk states.

- Through simulations and real world experiments on the Zerith humanoid robot, we demonstrate that our method exhibits strong robustness in agents and successfully validates sim-to-real transfer.

## 2 RELATED WORKS

**RL in Legged Locomotion Control** Reinforcement learning has become increasingly prevalent in the locomotion control of legged robots. In quadruped robotics, Lee et al. (2020); Cheng et al. (2024b); Fankhauser et al. (2018); Kumar et al. (2021); Nahrendra et al. (2023); Liu et al. (2024) employed an end-to-end proprioceptive-based training method for robust locomotion control, while Cheng et al. (2023); Agarwal et al. (2022); Zhuang et al. (2023); Hoeller et al. (2024) incorporated external perception for more complex and adaptable movements. Notably, He et al. (2024) implemented safety measures in the high-speed locomotion of quadruped robots, enabling highly flexible risk avoidance. As for humanoid robots, Reinforcement learning controllers are starting to demonstrate potential Siekmann et al. (2021b); Zhuang et al. (2024); Li et al. (2024); Radosavovic et al. (2024); Gu et al. (2024); Liao et al. (2024); Cheng et al. (2024a); Zhang et al. (2024). However, the stability of humanoid robots relies on bipedal balance control, which presents greater nonlinearity and complexity in locomotion control. This makes them more susceptible to external disturbances and internal errors, resulting in reduced fault tolerance. While many researchers are exploring how to push humanoid robots to perform extreme parkour, safety considerations in humanoid reinforcement learning controllers often remain unaddressed.

**Distributional Reinforcement Learning** Dist. RL have advanced considerably in recent years Bellemare et al. (2017); Dabney et al. (2018b;a); Yang et al. (2019). Different from traditional value function or action-value function learning methods, Dist. RL directly models the distribution of cumulative rewards. It starts from a probability perspective and considers the probability distribution of possible returns in a given state, rather than a single expected return value. Typically, these methods employ multiple quantile points to depict the return distribution and extend the Bellman equation into the Bellman distribution equation. These methods improve the performance of the policy more granularly by minimizing the distance between distributions. Dist. RL has not only achieved significant success in the Q-Learning framework, but has also been applied to the Actor-Critic architecture Nam et al. (2021); Barth-Maron et al. (2018); Duan et al. (2021), providing a new perspective for policy optimization and improving the robustness and decision-making of the policy.

**Dist. RL for Legged Locomotion Control** Many methods Tang et al. (2019); Stanko & Macek (2019); Shen et al. (2014); Théate & Ernst (2023) apply Dist. RL to train risk-sensitive policies. These methods train different policies by distorting the return distribution. Although Dist. RL has been applied in the real world Bellemare et al. (2020); Haarnoja et al. (2024), applying it to the field of motion control of humanoid robots is still a challenging task. Some methods Schneider et al. (2024); Shi et al. (2024); Tang et al. (2019) use the Actor-Critic architecture, model the value function as a Gaussian distribution, and use distorted expectations to optimize the worst-case policy, thereby improving the robustness of the agent's locomotion. These methods often employ a distorted risk measure with a fixed risk parameter, leading the agent to adopt an excessively cautious policy in some scenarios, which can impede the effectiveness of robot locomotion control. In addition, Dist. RL combined with a learnable perturbation module can also train robust locomotion policies Long et al. (2024).

## 3 METHOD

The overall architecture of HuRi is shown in Figure 2, where the Actor is responsible for outputting the actions of the humanoid robot, and the Critic outputs the probability distribution of the return. The risk distortion measure adjusts the agent's risk sensitivity by controlling the scalar risk parameter and reweighting the probability of possible outcomes. HuRi can adaptively adjust the risk parameter $\beta$ according to different environmental states to achieve a risk-aware policy. The following chapters will introduce each module in detail.

### 3.1 PRELIMINARY

**Theorem** We describe the locomotion problem of robots using a Partially Observable Markov Decision Process (POMDP) Shani et al. (2013); Spaan & Spaan (2004). The POMDP framework effectively models decision-making scenarios where information is incomplete, defining key elements such as states, actions, observations, and rewards. In this model, the environment at time step $t$ is represented by a complete state $s_t$. Based on the agent's policy, an action $a_t$ is performed,

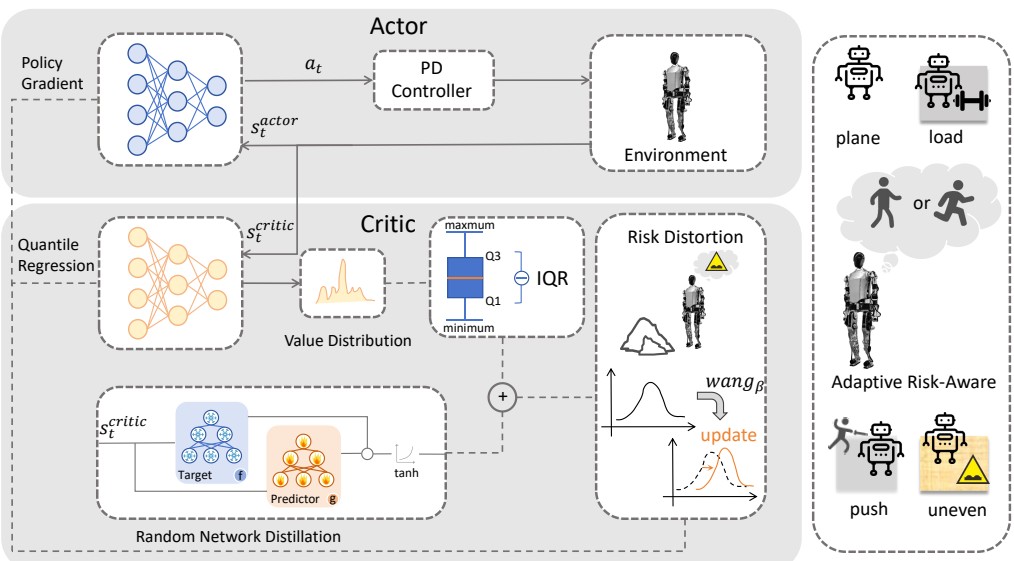

Figure 2: HuRi Architecture overview. The critic network is trained to estimate the distribution of returns, which is then utilized alongside a risk distortion metric to update the policy. HuRi uses IQR and RND to estimate the uncertainty in the environment and adaptively determine the scalar risk parameter. The image's right part illustrates the agent's capability to navigate various risk scenarios. Here, 'plane' denotes walking on flat terrain, 'load' refers to the robot's cargo, 'push' signifies sudden severe disturbances, and 'uneven' indicates traversing rough roads.

resulting in a state transition to $s_{t+1}$ with a probability $P(s_{t+1} \mid s_t, a_t)$. The agent then receives a reward $r_t$ and a partial observation $o_{t+1}$. The aim of reinforcement learning here is to identify a policy $\pi$ that maximizes the expected discounted sum of future rewards:

$$J(\pi) = \mathbb{E}_\pi \left[ \sum_{t=0}^\infty \gamma^t r_t \right] \tag{1}$$

**Action Space & State Space** We adopt asymmetric Actor-Critic structure as our training framework. The action space is $a_t \in \mathbb{R}^{12}$, representing the offset from the default position for each joint. The critic networks observe the global state $s_t^{critic} = [s_t^{actor}, v_t, h_t, e_t]$, which includes proprioceptive observations, the state space of actor $s_t^{actor}$, linear velocities $v_t$, feet surrounding height map $h_t$ and domain randomization variable $e_t$. For the actor networks, the state space contains only proprioceptive observations $s_t^{actor} = [\theta_t^{roll}, \theta_t^{pitch}, \omega_t^{roll}, \omega_t^{pitch}, c_t, q_t, \dot{q}_t, a_t]$, $\theta$ is the euler angle of robots' pelvis, $\omega$ is the angular velocity of orbots' pelvis; $c_t$ is the input command containing clock signal, desired linear velocity and angular velocity. $q$ and $\dot{q}$ represent position and velocity of each joint; $a_t$ represent the output action of policy.

**Actor-Critic Algorithm** The PPO algorithm, renowned for learning from interactions and regulating policy updates, has been chosen by HuRi for training sophisticated and unstable humanoid robots. HuRi's Actor aligns with the PPO, while the Critic incorporates the distributional reinforcement learning approach. It no longer outputs a scalar value $J_\pi$, but the entire distribution of the return $Z(s, a)$.

## 3.2 DISTRIBUTIONAL REINFORCEMENT LEARNING

As for distributional Critic, HuRi uses QR-DQN Dabney et al. (2018b) which uses quantized regression to approximate the return distribution. This probability distribution models the random variable $Z = \sum_{t=0}^\infty \gamma^t r_t$. In QR-DQN, the value distribution is parameterized as a set of quantiles $\{\theta_{\hat{\tau}_1}, \theta_{\hat{\tau}_2}, ..., \theta_{\hat{\tau}_n}\}$, which are predicted by the neural network and are the support points of the value

distribution. $\hat{\tau}_i = \frac{\hat{\tau}_{i-1} + \hat{\tau}_i}{2}$ for $1 \leq i \leq N$, where $\hat{\tau}_i = \frac{i}{N}$. In QR-DQN, the random return is approximated by a uniform mixture of N Diracs:

$$Z_\theta(s, a) := \frac{1}{N} \sum_{i=1}^{N} \delta_{\theta_i(s,a)} \tag{2}$$

Similar to ordinary reinforcement learning, Dist. RL uses a distributional Bellman operator to learn the entire action value distribution:

$$\mathcal{T}Z(s, a) :\overset{D}{=} R(s, a) + \gamma Z \left( X', \underset{a' \in \mathcal{A}}{\arg\max} \mathbb{E}[Z(S', a')] \right) \tag{3}$$

Where $:\overset{D}{=}$ means that two random variables have equal probability laws, and $S' \sim P(\cdot \mid s, a), A' \sim \pi(\cdot \mid s')$. The calculation of the distributional Bellman operator $\mathcal{T}Z(s, a)$ is based on the return distribution $Z$. The distributional Bellman operator is a contraction of p-Wasserstein Bellemare et al. (2017). Repeated application of the Bellman operator makes Dist. RL converge to the optimal policy during training.

HuRi uses $SR(\lambda)$ Nam et al. (2021) to calculate the target distribution. $SR(\lambda)$ generalizes the concept of the temporal difference (TD-$\lambda$) method to Dist. RL for calculating multi-step value targets. It generates target distribution $\mathcal{T}Z_\theta(s)$ by combining various distributions. In order to understand $SR(\lambda)$ more clearly, we give the process of $SR(\lambda)$ algorithm in the Algorithm.2. HuRi is similar to the method Schneider et al. (2024), using energy distance to measure the gap between the target distribution and the predicted critic distribution $Z_\theta(s)$ :

$$\mathcal{L}_{\text{quantiles}} = 2\mathbb{E}_{i,j}[\theta_i - \mathcal{T}\theta_j] - \mathbb{E}_{i,j}[\mathcal{T}\theta_i - \mathcal{T}\theta_j] - \mathbb{E}_{i,j}[\theta_i - \theta_j] \tag{4}$$

Equation (4) measures the difference between the target distribution and the predicted distribution through random sampling, where the distributions of $\theta$ and $\mathcal{T}\theta$ are derived from $Z_\theta$ and $\mathcal{T}Z_\theta$. Unlike this research Schneider et al. (2024), HuRi also uses MSE to measure the difference between the target expectation $J(\pi)$ and the expected $J_\beta(\pi)$ calculated by the probability distribution after implementing the risk distortion measure on the probability distribution. The calculation formula is as follows:

$$\mathcal{L}_{\text{expectation}} = MSE(E_{\tau \sim U[0,1]}[Z_\theta^{(\tau)}(S)], E_{\tau \sim U[0,1]}[Z_\theta^{\beta(\tau)}(S)]) \tag{5}$$

The expectation $E_{\tau \sim U[0,1]}$ in Equation (5) is computed over the $\tau$ values sampled from the uniform distribution $U[0, 1]$. HuRi uses the maximum PPO clip-objective to update the policy:

$$\mathcal{L}_{\text{surrogate}} = \min \left( \frac{\pi_\phi(s|a;r)}{\pi_{\phi_{\text{old}}}(s|a;r)} A^{\pi_{\phi_{\text{old}}}}(s, a; r), p\left(\epsilon, A^{\pi_{\phi_{\text{old}}}}(s, a; r)\right) \right)$$

$$\text{where} \quad p(\epsilon, A) = \begin{cases} (1+\epsilon)A, & \text{if } A \geq 0; \\ (1-\epsilon)A, & \text{if } A < 0. \end{cases} \tag{6}$$

### 3.3 Adaptive Risk-Aware Policy Learning

In the field of legged robot control, the policy of many methods Schneider et al. (2024); Shi et al. (2024); Tang et al. (2019) is to maximize the disorted expectation of value distribution. The distortion risk measure evaluates risk by re-weighting the probability of possible outcomes, typically reflecting the policy's preference for risk behavior. Unlike many previous methods that use CVaR to distort the distribution, HuRi uses the wang_function Wang (2000) to distort the value distribution. We calculate the quantile score of the distortion $h_\beta^{\text{Wang}}(\tau)$ as:

$$h_\beta^{\text{Wang}}(\tau) = \Phi(\Phi^{-1}(\tau) + \beta) \tag{7}$$

Where $\phi$ is the standard normal distribution and $\beta$ is the scalar risk parameter. In the remaining formulas, we abbreviate $h_\beta^{\text{Wang}}(\tau)$ to $\beta_{(\tau)}$. Wang_function adjusts the probability distribution in a nonlinear method. Compared with CVaR, wang_function has the ability to switch between risk-averse and risk-seeking policies. When $\beta = 0$, the policy is risk-neutral, when $\beta > 0$, the policy

is risk-averse, when $\beta < 0$, it is a risk-seeking policy. The scalar risk parameter $\beta$ can be considered a gauge of the agent's perception of risk, as a larger $\beta$ indicates a higher level of risk in the environment, necessitating a more conservative approach to policy. Therefore, $\beta$ represents the risk sensitivity of the agent, which is very important for the success of training. A survey Schubert et al. (2021) has proved that it is suboptimal to adopt a fixed risk sensitivity in a dynamic risk environment. Excessively cautious behavior hinders the thorough exploration needed during agent training, while overly adventurous behavior can result in a higher frequency of falls throughout the training process. For this reason, HuRi proposed a method to adaptively adjust the risk sensitivity according to the current state of the agent, allowing the agent to take cautious behavior in the risky environment conditions and take exploratory behavior after being more familiar with the environment.

**Inter Quartile Range Module** A previous research Dabney et al. (2018a) defines risk as the uncertainty of possible outcomes, and divides uncertainty into intrinsic uncertainty and parameter uncertainty. Intrinsic uncertainty refers to the uncertainty of the environment itself, which cannot be eliminated even if the agent has a perfect understanding of the environment. Parameter uncertainty is typically associated with Bayesian reinforcement learning, which refers to the uncertainty of the parameters of the environmental model (such as transition probabilities and reward functions). Parameter uncertainty reflects the incompleteness of the agent's cognition of the environment, that is, the uncertainty of the agent in its predicted environment and rewards. The probability distribution obtained by Dist. RL is mainly used to capture intrinsic uncertainty. HuRi uses the interquartile range (IQR) to measure intrinsic uncertainty:

$$IQR = Q_3 - Q_1, \quad Q_3 = F_Z^{-1}(0.75), \quad Q_1 = F_Z^{-1}(0.25). \tag{8}$$

HuRi sets a threshold range of intrinsic uncertainty $[t_{min}, t_{max}]$. When $IQR > t_{max}$, it means that there is strong intrinsic uncertainty in the current environment, and the agent needs to adopt a more cautious policy. We set the risk parameter $\beta_{IQR} = 1$. Similarly, $IQR \in [t_{min}, t_{max}]$ is to adopt a risk-neutral policy $\beta_{IQR} = 0$; when $IQR < t_{min}$, $\beta_{IQR} = -1$, and a risk-seeking policy is adopted to increase exploration during training.

**Random Network Distillation Module** It is not comprehensive to use only IQR to measure intrinsic uncertainty to approximate the environmental risk level. HuRi uses random network distillation(RND) Burda et al. (2018) to measure parameter uncertainty in the environment and further approximate the actual risk level in the environment. RND uses a frozen randomly initialized neural network (target network) $g$ and a trainable neural network (predictor network) $f$. The parameters of the target network are fixed during training, and the predictor network is trained to imitate the output of the target network as much as possible. The random network distillation method uses MSE to reduce the prediction error:

$$Loss_{RND}(s_t^{critic}) = \left(f(s_t^{critic}) - g(s_t^{critic})\right)^2 \tag{9}$$

The prediction error can evaluate the uncertainty in the dynamic environment conditions. HuRi's assessment of parameter uncertainty further corrects the scalar risk parameter $\beta$ in the distortion function. RND reflects the agent's familiarity with the state during training. If there are multiple unknown states in a given environment, the agent should adopt a risk-averse policy, which is conducive to the agent's safe exploration. If there is a significant difference between the output of the predictor network and the target network, indicating that the environment is relatively novel for the agent and the possibility of robot falling increases. Therefore, the agent should increase its risk sensitivity. We define the relationship between the scalar risk parameter and the RND loss:

$$\beta_{RND} = tanh(Loss_{RND}) \tag{10}$$

The calculation formula for measuring the scalar risk parameter by combining intrinsic uncertainty and parameter uncertainty is as follows:

$$\beta = \beta_{IQR} + \beta_{RND} \tag{11}$$

### 3.4 Loss Function

The calculation formula of HuRi's overall loss function is

$$\mathcal{L} = \mathcal{L}_{\text{surrogate}} + \lambda_{\text{expectation}} \cdot \mathcal{L}_{\text{expectation}} + \lambda_{\text{quantiles}} \cdot \mathcal{L}_{\text{quantiles}} + \lambda_{\text{entropy}} \cdot \mathcal{L}_{\text{entropy}} \tag{12}$$

Among them, $\mathcal{L}_{\text{quantiles}}$ calculates the quantile loss, and uses the quantile energy loss for calculation to measure the difference between distributions. Unlike other Dist. RL, Huri also used MSE loss to the distorted expectations. MSE provides additional information about the predicted distribution as the second-order moment of the prediction error. In addition, the use of $\mathcal{L}_{\text{entropy}}$ in our training process helps to maintain diversity and exploration in the policy.

# 4 EXPERIMENTS

## 4.1 EXPERIMENTS SETTING

**Benchmark Comparision.** For a comparative evaluation, the experiments we performed are as follows:

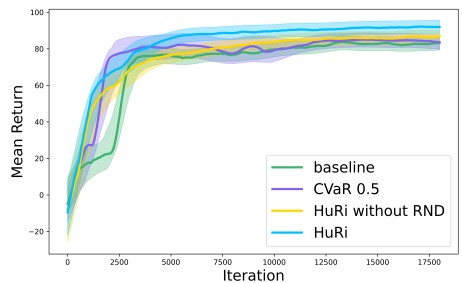

- **Baseline**: Train the policy using original PPO.

- **Cvar0.5**: Employ CVAR as distortion function, and risk parameter is 0.5. Use the same hyperparameters and loss function as huri.

- **HuRi w/o RND**: Our method without RND. The rest is consistent with huri.

**Training setting:** All experiments are training on plane terrain in the Isaac Gym, with 4096 Zerith-1 environments in parallel. All methods

Figure 3: Reward Comparison: The agent's actual return during training is shown in the figure, where the thick line represents the average return, and the shaded regions indicate the 95% confidence intervals across different seeds. HuRi achieves the highest convergent reward.

have same hidden layer dimension with [512, 256, 128]. Specifically, the Critic of Huri outputs calculated values of 64 quantiles. During training and deployment, we employed PD position controllers for each join. All the reward function are detailed in Appendix A.A.2. It costs 18 hours for each method traning and about 18000 iterations, utilizing a single NVIDIA RTX 4090 with 24 GB memory.

## 4.2 SIMULATION EXPERIMENTS

We conducted experiments with five random seeds, training each seed five times, and the results are shown in the Figure 3. It is obvious that our method(average return 90.86) better than baseline(83.28), CVaR0.5(84.17), HuRi w/o RND(86.93). We believe that HuRi can adaptively adjust the risk sensitivity of its policy in dynamic environments, deeply exploring and selecting optimal actions during training to achieve higher rewards. However, high rewards do not necessarily indicate strong resilience to risk. To further verify HuRi's robustness in motion control, we considered various risk factors, including sustained external forces, sudden impacts, and load variations, etc.

The first experimental settings involved applying random continuous disturbances to the humanoid robot's centroid, feet, and hands. These disturbances were sampled from a uniform distribution between 0 and 100 N, changing every 5 steps. It is worth noting that the range of external disturbances during training is [0,10] N, and these disturbances are applied solely to the centroid. The range of disturbances during testing was far beyond the range of the training settings. Details on the domain randomization parameters can be found in the Appendix A.5. In the second experiment, we applied sudden impacts to the same areas of the robot, with forces sampled from a uniform distribution ranging from 150 N to 200 N, delivered every 2 seconds. The robot was commanded to move at a constant speed of 1 m/s, which exceeded its training maximum of 0.7 m/s. Any falls during its walk were classified as failures. We recorded the success rate of the robot for each trial. To reduce variability, we used five different random seeds, with each seed repeated 10 times. Table 1 presents the final results, showing that HuRi demonstrated superior performance in handling continuous external disturbances and sudden impacts on the centroid, hands, and feet.

|  | Continuous disturbances | | | Sudden extreme disturbances | | |
|---|---|---|---|---|---|---|
|  | centroid | hand | feet | centroid | hand | feet |
| **baseline** | 0.6657 | 0.6178 | 0.6583 | 0.5750 | 0.5933 | 0.5886 |
| **CVaR 0.5** | 0.6870 | 0.6411 | 0.6981 | 0.6092 | 0.6267 | 0.6267 |
| **HuRi w/o RND** | 0.8186 | 0.7700 | 0.8482 | 0.7758 | 0.8078 | 0.8077 |
| **HuRi** | **0.8562** | **0.8090** | **0.8658** | **0.8317** | **0.8116** | **0.8171** |

Table 1: Comparison of success rate under different disturbances. We perform continuous and sudden extreme disturbances on the robot's hands, legs, and centroid, respectively. If the robot falls, it is considered a failure.

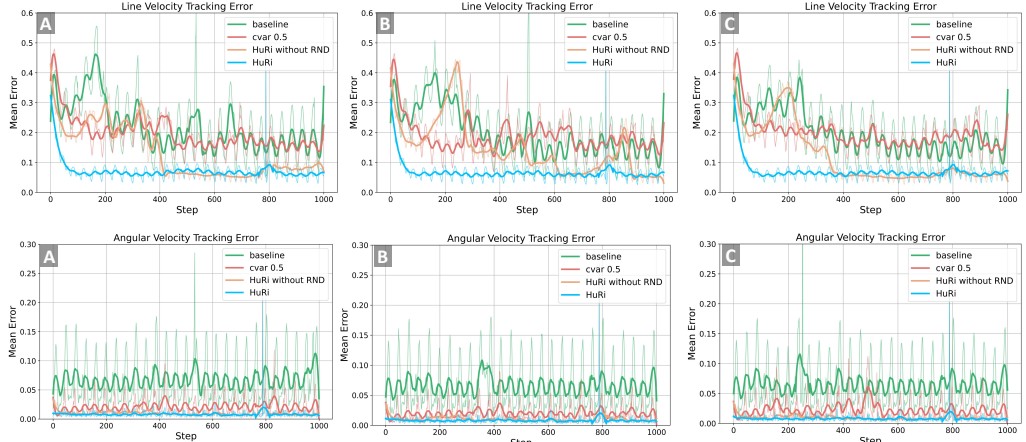

Figure 4: Error Comparison: Velocity tracking error under different disturbances. The top image shows the linear velocity error, while the bottom image represents the angular velocity error. A represents load disturbances, B represents friction disturbances, and C represents both disturbances. HuRi has the lowest velocity tracking error.

To further demonstrate the effectiveness of HuRi's adaptive risk-aware ability, we designed three sets of experiments. In the first set(Figure 4.A), we varied the robot's load. In the second set of experiments(Figure 4.B), we altered the ground friction. The third set of experiments(Figure 4.C) combined both load and friction disturbances to examine whether Huri can handle more complex risk scenarios. For all three sets of experiments, the robot's speed was set to 1 m/s and the angular velocity to 0, with a random external force sampled from a uniform distribution of [0, 100] N applied every 0.5 second. The range of disturbances during testing was far beyond the range of the training settings. Details on the domain randomization parameters can be found in the Appendix A.5. We randomly selected four seeds, simulated 1024 environments in parallel, and averaged the experimental results. The Figure 4 showcases the tracking errors for both the average linear velocity and angular velocity across the three experiments. We found that HuRi's velocity errors were significantly smaller than those of the other three methods. HuRi maintained highly robust performance amidst diverse disturbances, indicating that HuRi thoroughly explored the potential risk factors affecting the agent during training.

Additionally, we sought to demonstrate through experiments that HuRi's estimation of risk levels is relatively accurate. We tracked the scalar risk parameter $\beta$ and value distributions during three scenarios: the robot's normal walking on plane terrain, exposure to a 200N sudden extreme disturbance, and traversal on uneven terrain. The results are shown in the Figure 5. Notably, due to our method was trained on plane terrain, it is intuitive to expect that walking on uneven terrain presents the highest risk for the robot. The cumulative distribution function in Figure 5.A clearly shows that the rewards on uneven terrain are significantly lower than the other two scenarios, indicating a higher likelihood of robot falls.

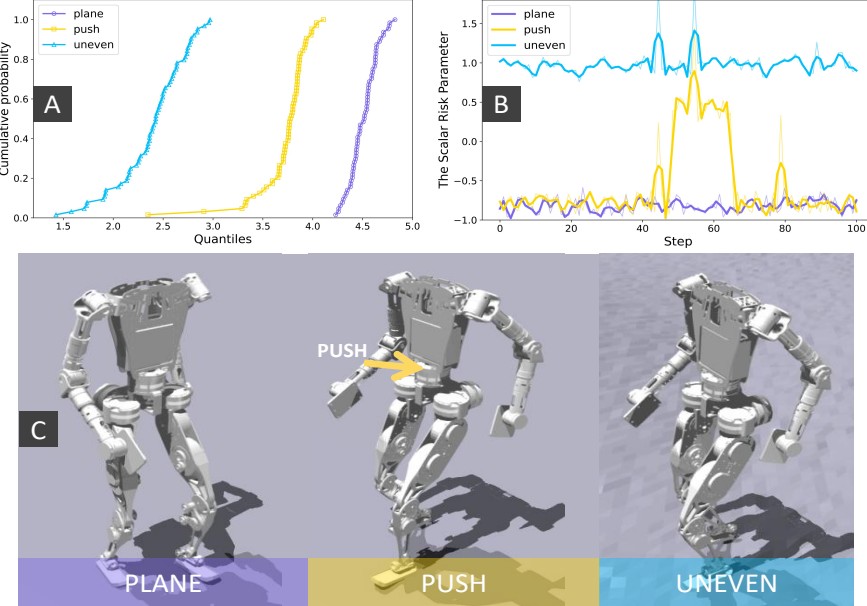

Figure 5: Figure A displays the variance in value distribution produced by the Critic under various risk scenarios. The horizontal axis is the predicted quantile and the vertical axis is the cumulative distribution probability value. Figure B shows the change of the scalar risk parameter beta of the distortion metric. Figure C shows the situation of the robot walking on flat ground, suffering sudden extreme disturbance, and walking on a rough road in the simulation environment.

Through quantitative analysis, we observed that IQR(uneven) > IQR(push) > IQR(plane), indicating that the intrinsic uncertainty assessed by IQR aligns with the actual environment conditions. Figure 5.B visually demonstrates that the robot adopted an extremely cautious policy when navigating the previously untrained uneven terrain. In contrast, when subjected to sudden extreme disturbance on flat ground, the scalar risk parameter $\beta$ sharply increased, indicating that HuRi can achieve robust motion control in high-risk scenarios.

### 4.3 REAL WORLD EXPERIMENTS

Domain randomization is used in training to reduce the sim-to-real gap by simulating diverse environments. This involves randomizing dynamic parameters such as body mass and ground friction in each episode, etc. Additionally, random forces are applied to the robot, and sensor feedback is noisy to enhance the controller's resilience to measurement errors and faults. The specific parameters for randomization are listed in Table 5. In the real-world experiments, we primarily measured the impact of disturbances on the robot's stability. These disturbances included additional loads on the centroid, extra loads on the end effectors, and external pulling forces, etc.

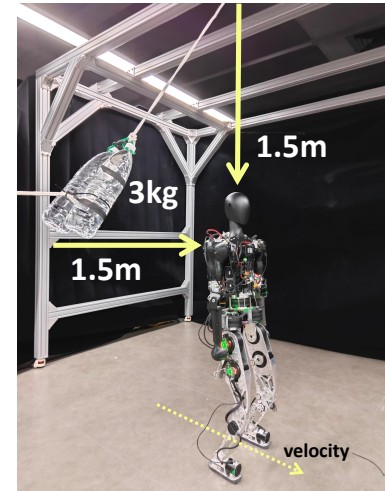

Figure 6: Diagram of the pendulum system experimental setup

Firstly, a fixed lateral impact force is applied to the robot using a pendulum system. The pendulum has a height of 1.5 meters, with the weight released from a fixed angle at a horizontal distance of 1.5 meters from the pivot point. The experimental setup is shown in Figure 6. At the lowest point of its swing, the weight strikes the side of the robot, generating a constant external force. A 3 kg water bottle is used as the pendulum's weight. The robot's success rate of surviving under lateral impact is evaluated at a speed of 0.6 m/s. Subsequently, we measured the velocity error rate under

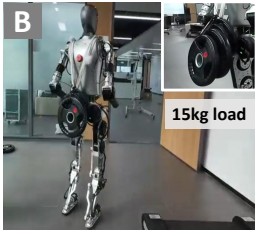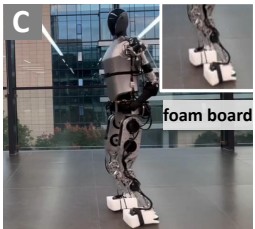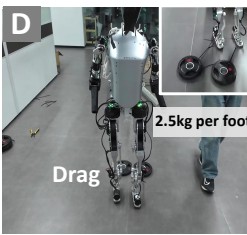

Figure 7: Real-World Experiments: (A) Walk on uneven terrain. (B) A 15 kg load is added to the centroid. (C) White foam board insoles are placed under the feet. (D) A 2.5 kg load is added to each foot. In all these scenarios, our method demonstrates robust performance.

additional loads applied at the centroid or the feet. During the experiment, a 5 kg load was added to the robot's centroid, and an additional 3 kg load was placed on each foot. The latter load generated a significant torque at the robot's thigh joint. The tests were conducted at velocities of 0.3 m/s, 0.6 m/s, and 0.9 m/s, with the experimental results shown in Table 2.

| | **External Force** success rate% | **Centroid Load** velocity error rate% | | | **Centroid Load** velocity error rate% | | |
|---|---|---|---|---|---|---|---|
| **velocity** | **0.6 m/s** | **0.3 m/s** | **0.6 m/s** | **0.9 m/s** | **0.3 m/s** | **0.3 m/s** | **0.9 m/s** |
| **baseline** | 35 (7/20) | 24.2 | 28.3 | 29.5 | 36.8 | 31.6 | 37.1 |
| **CVaR 0.5** | 40 (8/20) | 20.8 | 23.7 | 24.8 | 27.3 | 20.4 | 33.4 |
| **HuRi w/o RND** | 55 (11/20) | 12.6 | 13.3 | 19.7 | 12.3 | 17.6 | 30.5 |
| **HuRi** | **65 (13/20)** | **7.3** | **5.6** | **12.3** | **9.3** | **11.7** | **20.2** |

Table 2: In the real-world experiments, when the robot was subjected to external forces, our method achieved the highest success rate. In experiments where additional loads were applied to the centroid of robot or the feet, we assessed the velocity error rate. Under various velocity commands, our approach consistently resulted in the lowest velocity error.

Experimental results demonstrate that our approach effectively resists out-of-distribution disturbances, showcasing safe and robust motion control capabilities. During testing, we observed that even with an additional 15 kg load at the robot's center of mass (approximately 42% of the robot's body weight), our method was still able to maintain stable movement and standing. Furthermore, we tested our approach on surfaces with varying friction coefficients by changing the robot's insoles. The results in Figure indicate that our method remains robust and capable of walking stably across different frictional surfaces. For further real-world experimental details, please refer to the supplementary video.

## 5 CONCLUSION, LIMITATIONS AND FUTURE DIRECTIONS

In this work, we proposed an adaptive risk-aware distributional reinforcement algorithm. By adaptively adjusting the agent's sensitivity to risk according to the environmental risk assessment, the agent can thoroughly explore the various uncertainties present during training. This enables the robot to withstand diverse external interferences and achieve a robust locomotion control policy. Simulations and physical experiments indicate that HuRi can equip robots with the ability to withstand various interferences. Since our method is based on the traditional PPO algorithm without relying on historical information, our approach is inferior to the latest research on locomotion control on multiple terrains. In the future, we will focus on how to improve the robustness of humanoid robot motion control on multiple terrains.

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

# A APPENDIX

## A.1 HYPERPARAMETERS OF HURI

In the training phase, we configured the hidden dimensions of the Actor and Critic networks across all models to [512, 256, 128], established the Actor's input dimension at 46, set the Critic's input dimension to 399, and determined the output quantiles dimension to be 64. In PPO, the coefficient $\gamma$ used for calculating the discounted reward is 0.9, the clip parameter is fixed at 0.2, and the learning rate is set to 2e-4. When $SR(\lambda)$ calculates the target distribution, $\lambda = 1$. The hyperparameters are listed in Table3.

| Hyperparameter | Value |
|---|---|
| Iterations | 18000 |
| Hidden State | [512, 256, 128] |
| $\lambda_{expectation}$ | 0.05 |
| $\lambda_{quantiles}$ | 1.0 |
| $\lambda_{entropy}$ | 0.01 |
| Iterations | 18000 |
| IQR Range | [0.3, 0.7] |
| Discount Factor | 0.99 |
| GAE Parameter | 0.95 |
| Timesteps per Rollout | 60 |
| Epochs per Rollout | 8 |
| Minibatches per Epoch | 4 |
| Entropy Bonus ($\alpha_2$) | 0.01 |
| Value Loss Coefficient ($\alpha_1$) | 1.0 |
| Clip Range | 0.2 |
| Reward Normalization | yes |
| Learning Rate | $2e-4$ |
| # Environments | 4096 |
| Optimizer | Adam |
| RND Leanring Rate | 1e-3 |
| RND Hidden State(g) | [32, 32] |
| RND Hidden State(f) | [32] |
| RND optimizer | Adam |

Table 3: HuRi hyperparameters.

## A.2 TRAINING DETAILS

We used the reward function as shown in Table 4, where the task reward guides the robot to track the desired speed and complete motions on various terrains and alive reward mitigates the exploration burden in early period. Besides, we design comprehensive reward about feet (Siekmann et al. (2021a),Margolis & Agrawal (2023)) to guide locomotion and prevent weird posture. Through extensive training trials, we optimized our reward weight settings to ensure that the robot moves in a relatively ideal manner.

## A.3 DOMAIN RANDOMIZATION PARAMETERS IN TRAINING AND TESTING

The range of disturbances during testing was far beyond the range of the training settings. Parameters are shown in Table 5.

| Term | Equation | Weight |
|------|----------|--------|
| **Task Reward** | | |
| alive | 1 | 0.5 |
| xy velocity tracking | $\exp\{-\|\mathbf{v}_{xy} - \mathbf{v}_{xy}^{\text{cmd}}\|^2 * 5\}$ | 1.5 |
| yaw velocity tracking | $\exp\{-(\boldsymbol{\omega}_z - \omega_z^{\text{cmd}})^2 * 5\}$ | 1.0 |
| **Feet Guidance** | | |
| swing phase tracking (force) | $\sum_{\text{foot}}[1 - C_{\text{foot}}^{\text{cmd}}(\boldsymbol{\theta}^{\text{cmd}}, t)]\exp\{-\|\mathbf{f}^{\text{foot}}\|^2/100\}$ | 5.0 |
| stance phase tracking (velocity) | $\sum_{\text{foot}}[C_{\text{foot}}^{\text{cmd}}(\boldsymbol{\theta}^{\text{cmd}}, t)]\exp\{-\|\mathbf{v}_{xy}^{\text{foot}}\|^2/5\}$ | 10.0 |
| raibert heuristic footswing tracking | $(\mathbf{p}_{x,y,\text{foot}}^{f} - \mathbf{p}_{x,y,\text{foot}}^{f,\text{cmd}}(\boldsymbol{s}_y^{\text{cmd}}))^2$ | $-30.0$ |
| footswing height tracking | $\sum_{\text{foot}}(\boldsymbol{h}_{z,\text{foot}}^{f} - \boldsymbol{h}_z^{f,\text{cmd}})^2 C_{\text{foot}}^{\text{cmd}}(\boldsymbol{\theta}^{\text{cmd}}, t)$ | $-10.0$ |
| **Regularization Reward** | | |
| body height | $\exp\{-(\boldsymbol{h_z} - \boldsymbol{h}_z^{\text{cmd}})^2 * 1000\}$ | $-0.2$ |
| z velocity | $\mathbf{v}_z^2$ | -0.02 |
| foot slip | $\|\mathbf{v}_{xy}^{\text{foot}}\|^2$ | -0.04 |
| hip position | $\exp\{-\sum_{i=1}^{2} q_{roll,yaw}^2 * 100\}$ | 0.4 |
| feet orientation | $\exp\{-\sum_{i=1}^{2} \|\theta_{roll,pitch}^{\text{foot}}\| * 10\}$ | 0.4 |
| feet stumble | $\mathbb{1}(\max_i(\sqrt{F_{x_i}^2 + F_{y_i}^2} > 4\|F_{z_i}\|))$ | -1 |
| orientation | $\exp\{-\|g_{xy}\|^2 * 10\}$ | 1.5 |
| thigh/calf collision | $1_{\text{collision}}$ | $-5.0$ |
| joint limit violation | $1_{q_i > q_{max} \|\| q_i < q_{min}}$ | $-10.0$ |
| joint torques | $\|\boldsymbol{\tau}\|^2$ | -1e-5 |
| joint velocities | $\|\dot{\mathbf{q}}\|^2$ | -1e-3 |
| joint accelerations | $\|\ddot{\mathbf{q}}\|^2$ | -2.5e-7 |
| action rate | $\|\mathbf{a}_t\|$ | -5e-5 |
| action smoothing | $\|\mathbf{a}_{t-1} - \mathbf{a}_t\|^2$ | -0.01 |
| action smoothing, 2nd order | $\|\mathbf{a}_{t-2} - 2\mathbf{a}_{t-1} + \mathbf{a}_t\|^2$ | -0.01 |

Table 4: Reward structure

Table 5: Domain randomization parameters in training and testing

| Parameters | Range in Training [Min, Max] | Range in Testing [Min, Max] |
|-----------|------------------------------|------------------------------|
| forces on centroid | [0, 10] N | [0, 100] N for continuous disturbances |
| forces on centroid | [0, 10] N | [150, 200] N for sudden extreme disturbances |
| forces on hands | 0 N | [0, 100] N for continuous disturbances |
| forces on hands | 0 N | [150, 200] N for sudden extreme disturbances |
| forces on feet | 0 N | [0, 100] N for continuous disturbances |
| forces on feet | 0 N | [150, 200] N for sudden extreme disturbances |
| line velocity | [0, 0.7] m/s | 1 m/s |
| mass disturbances | [-2, 5] kg | [-3, 8] kg |
| friction disturbances | [0.1, 1.5] | [0.1, 2] |
| body com | [-0.07, 0.1] kg | [-0.07, 0.1] kg |

## A.4 ALGORITHM

We employ algorithmic blocks to delineate the detailed flow of the algorithm. The algorithm of Huri is shown in the Algorithm.1. The process of the SR($\lambda$) algorithm is shown in the Algorithm.2.

Table 6: Other Domain Randomizations

| Parameter | Range [Min, Max] |
|---|---|
| Link Mass | $[-0.8, 1.4] \times$ default kg |
| Base Orientation Roll Pitch | $[-0.1, 0.1], [-0.1, 0.1] \times$ rad |
| Motor Strength | $[0.9, 1.1] \times$ default Nm |
| Joint Kp | $[0.85, 1.15] \times$ default |
| Joint Kd | $[0.85, 1.15] \times$ default |
| Initial Joint Positions | $[0.5, 1.5] \times$ default |
| System Delay | $[0, 40]$ ms |
| Push Velocity XY | $[0, 0.5]$ m/s |

---

**Algorithm 1** HuRi Adaptive Risk-Aware Reinforcement Learning

---

**Require:** Initial environment state $s_0$
**Ensure:** Optimal action policy $\pi^*$
 1: Initialize actor-critic networks with parameters $\psi$ and $\phi$
 2: Set IQR thresholds $t_{min}, t_{max}$
 3: Initialize RND networks: target network $g$ and predictor network $f$
 4: **for** each episode **do**
 5:     Reset environment to initial state $s_0$
 6:     **for** each timestep $t$ **do**
 7:         Observe current state $s_t$
 8:         Actor selects action $a_t$ based on policy $\pi$ parameterized by $\psi$
 9:         Execute action $a_t$ in environment
10:         Observe reward $r_t$ and new state $s_{t+1}$
11:         Estimate return distribution $Z_\theta(s_t, a_t)$ using critic
12:         Calculate intrinsic uncertainty using IQR: $IQR = Q_3 - Q_1$
13:         **if** $IQR > t_{max}$ **then**
14:             $\beta_{IQR} \leftarrow 1$                                                 ▷ Risk-averse policy
15:         **else if** $t_{min} \leq IQR \leq t_{max}$ **then**
16:             $\beta_{IQR} \leftarrow 0$                                                ▷ Risk-neutral policy
17:         **else**
18:             $\beta_{IQR} \leftarrow -1$                                             ▷ Risk-seeking policy
19:         **end if**
20:         Compute $Loss_{RND} \leftarrow \left(f(s_t^{critic}) - g(s_t^{critic})\right)^2$
21:         Set parameter uncertainty risk parameter $\beta_{RND} \leftarrow \tanh(Loss_{RND})$
22:         Calculate overall risk parameter $\beta \leftarrow \beta_{IQR} + \beta_{RND}$
23:         Adjust return distribution using Wang distortion function: $h_\beta^{Wang}(\tau) = \Phi(\Phi^{-1}(\tau) + \beta)$
24:         Compute the expected return $E[Z_\theta(s_t, a_t)]$ using the distorted value distribution:
25:         $E[Z_\theta(s_t, a_t)] == \int_0^1 h_\beta^{Wang}(\tau) Z_\theta^\tau(s) \, d\tau$
26:         Calculate loss for value distribution $L_{quantiles}$
27:         Calculate expectation loss $L_{expectation}$ using MSE
28:         Update critic network parameters $\phi$ to minimize:
29:         $L \leftarrow \lambda_{expectation} \cdot L_{expectation} + \lambda_{quantiles} \cdot L_{quantiles}$
30:         Update actor network parameters $\psi$ using PPO to maximize policy objective
31:     **end for**
32: **end for**
33: **return** Optimal policy $\pi^*$

---

**Algorithm 2** SR($\lambda$)

---

**Require:** Transition samples $(s_t, a_t, r_{t+1}, s_{t+1})$, current value distribution parameters $\theta$, discount factor $\gamma$, eligibility trace decay parameter $\lambda$
1: Initialize eligibility traces $e(s) = 0$ for all states $s$
2: **for** each time step $t$ **do**
3:      Observe transition $(s_t, a_t, r_{t+1}, s_{t+1})$
4:      Compute TD error: $\delta_t = r_{t+1} + \gamma Z_\theta(s_{t+1}) - Z_\theta(s_t)$
5:      Update eligibility trace for state $s_t$: $e(s_t) = e(s_t) + 1$
6:      **for** each state $s$ **do**
7:          $Z_\theta(s) \leftarrow Z_\theta(s) + \alpha \delta_t e(s)$
8:          Update the eligibility trace: $e(s) \leftarrow \gamma \lambda e(s)$
9:      **end for**
10: **end for**

---

### A.5 Ablation Experiments

To further validate the contribution of each module in HuRi, we conducted the following ablation experiments:

- **HuRi w/o RND**: Our method without the RND module.

- **HuRi w/o IQR**: Our method without the IQR module.

We conducted experiments with five random seeds, training each seed five times, and the results are shown in the Figure 8. In the Method section 3, we explained that IQR is used to measure intrinsic uncertainty, while RND quantifies parameter uncertainty. Combining these two uncertainties to assess the risk level in the environment aids in safe exploration for the agent,

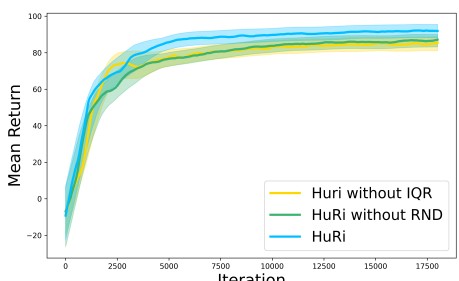

Figure 8: The agent's actual return during training is shown in figure, where the thick line represents the average return, and the shaded regions indicate the 95% confidence intervals across different seeds. HuRi achieves the highest convergent reward.

improving its rewards. According to the experimental settings in the section 4.2, we applied continuous disturbances and extreme sudden disturbances to the agent's centroid, hands, and feet, and the results are shown in Table 7. Additionally, following parameter settings of another simulation experiments, we applied mass disturbances, friction disturbances, and both types of disturbances to the agent, tracing the velocity error. The experimental results are shown in the Figure 9.

|  | **Continuous disturbances** | | | **Sudden extreme disturbances** | | |
|---|---|---|---|---|---|---|
|  | **centroid** | **hand** | **feet** | **centroid** | **hand** | **feet** |
| **HuRi w/o IQR** | 0.8102 | 0.8037 | 0.8283 | 0.7894 | 0.7995 | 0.7868 |
| **HuRi w/o RND** | 0.8186 | 0.7700 | 0.8482 | 0.7758 | 0.8078 | 0.8077 |
| **HuRi** | **0.8562** | **0.8090** | **0.8658** | **0.8317** | **0.8116** | **0.8171** |

Table 7: Comparison of success rate under different disturbances. We perform continuous and sudden extreme disturbances on the robot's. HuRi demonstrates the most effective resistance to various disturbances.

The results indicate that our method, HuRi, achieved the best performance. Without the RND module in our method, the value of $\beta$ can only switch between -1, 0, and 1, which fails to accurately estimate the risk level in the environment and is insufficient to handle the complex changes in varying environmental conditions. On the other hand, without the IQR module, since $\beta_{RND}$ is greater than or equal to 0, the agent cannot switch to a risk-seeking policy. This results in the agent consistently choosing lower-risk actions, hindering exploration during the training process and reducing overall adaptability and performance.

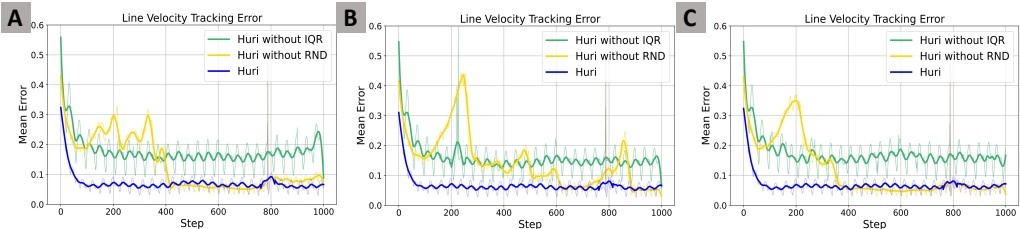

Figure 9: Error Comparison: The figure shows the linear velocity error. A represents load distur-bances, B represents friction disturbances, and C represents both disturbances. HuRi has the lowest velocity tracking error.

Through ablation experiments, we validated the contribution of each module and theoretically an-alyzed the shortcomings of using the IQR and RND modules individually. The structural design of HuRi integrates the advantages of both modules from the perspective of combining two types of uncertainty, while avoiding the drawbacks of each, thereby achieving the best experimental results.

## A.6 Verification of Model Independence from Reward Formulation

To demonstrate that HuRi does not rely on specific robots and reward formulations, we conducted training and testing on the Unitree Go2 quadruped robot, comparing the baseline with HuRi. The training consisted of 2048 environmental instances, while other settings remained consistent with those described in Rudin et al. (2022). We performed five experimental repetitions using five random seeds. The training results, as illustrated in Figure 10, indicate that our method enables the robot to traverse diverse terrains while achieving higher rewards.

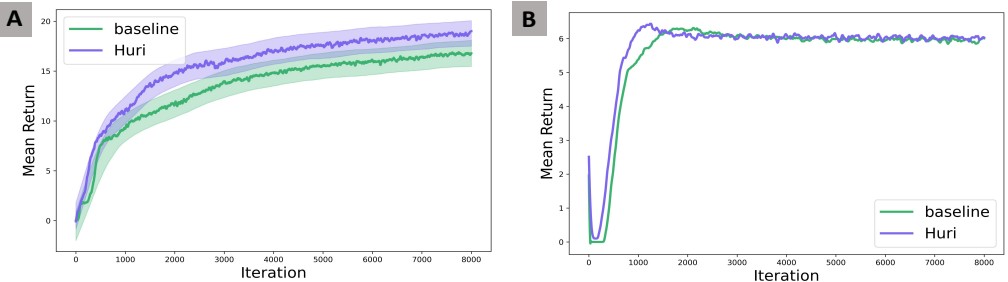

Figure 10: The agent's actual return during training is shown in figure A, where the thick line represents the average return, and the shaded regions indicate the 95% confidence intervals across different seeds. HuRi achieves the highest convergent reward. Figure B illustrates the variation in terrain level throughout the training process.

To verify the effectiveness of our method un-der varying reward formulations, we measured the robot's success rate in high platforms under various perturbations. During the testing phase, we applied external forces to the robot's cen-troid randomly every 100 steps within a range of [0, 100] N and mass disturbances within [-1, 1] kg. These perturbations were beyond the range encountered during training. The results

| Height | Policy | Success % |
|--------|--------|-----------|
| 0.4m   | baseline | 65.37 |
|        | HuRi     | 92.73 |
| 0.45m  | baseline | 39.01 |
|        | HuRi     | 80.55 |

Table 8: Success rates of robot walking down a platform.

summarized in Table 8, demonstrate that our method achieves robust performance even under these challenging environments. The experiments also demonstrated that our method is not dependent on specific reward formulations and possesses good generalization performance.

## A.7 Training and Testing on Various Terrains

To verify that the risk preference of our method does not negatively impact the agent's mobility, we trained the four methods from the paper(baseline, cvar0.5, HuRi without RND, and HuRi) on multiple terrains, including 'rough slope up', 'rough slope down', and 'discrete'. The other training settings were consistent with those in the paper.

| Policy | Success Rate % |
|---|---|
| baseline | 25.14 |
| CvaR 0.5 | 49.78 |
| HuRi w/o RND | 46.04 |
| HuRi | 57.76 |

Table 9: Success rates of robot walking through through multiple terrains.

The training results are shown in the figure 11. Considering both the rewards and the terrain levels, our method achieved the best performance. To further test the robustness of HuRi's motion control across multiple terrains, we randomly applied external force to the robot's centroid disturbances in the range of [0, 100]N in a multi-terrain environment, selected four random seeds, and tested in parallel across 1024 environments. The test results are presented in Table 9.

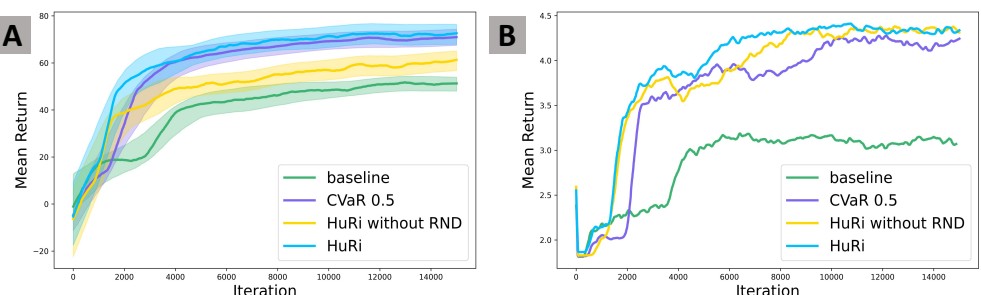

Figure 11: Figure A shows the agent's return during training, with the thick line representing the average return and the shaded regions indicating 95% confidence intervals across different seeds. HuRi achieves the highest reward. Figure B depicts the terrain level variation during training."

The experimental results demonstrate that our method achieves superior performance. Combined with the above experiments on the quadruped robot, we conclude that HuRi enhances the robustness of robotic motion control.

### A.8 FAILURE OF HuRi WITH THE ALTERED DISTORTION FUNCTION

The distortion function(wang_function Wang (2000)) also plays a role in HuRi. To demonstrate the compatibility of wang_function with HuRi, we trained the model using the CVaR distortion function combined with the IQR and RND modules. CVaR focuses on the tail of the distribution, emphasizing the lower tail (risk-averse) or the upper tail (theoretically risk-seeking). When $\beta < 1$, the CVaR function only considers the outcomes below a certain quantile, ignoring the rest of the distribution. This design is particularly suitable for emphasizing unfavorable outcomes to mitigate risk. However, to adjust for risk-seeking behavior, attention must be directed to the upper tail of the distribution, which mathematically requires $\beta > 1$. At this point, CVaR extends beyond its domain (e.g., expanding the sampling range to $[0, \beta]$), leading to practical difficulties. Therefore, policies based on CVaR can only be risk-neutral or risk-averse. We set different IQR thresholds $t$; if $IQR > t$, then $\beta_{IQR} = 0.5$, otherwise, $\beta_{IQR} = 1$, with all other settings consistent with HuRi. The training results are shown in the Figure 12.

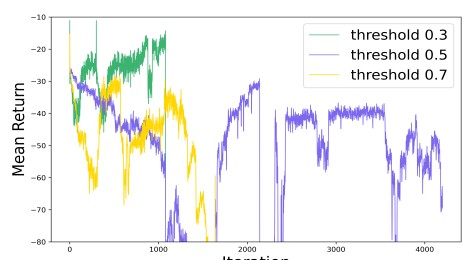

Figure 12: Reward curve using CVaR distortion function: The yellow and green curves vanish in the second half of the figure due to rewards falling below -80.

We tested multiple thresholds, and the failure of the training results showed that the CVaR distortion function could not effectively integrate with the IQR and RND modules. There are mainly the following reasons. Firstly, the CVaR distortion function is inherently a linear distortion, and its

linear adjustments to the tail of the distribution do not align well with the complex nonlinear relationships of the IQR and RND modules. The IQR and RND modules are better suited for capturing the complex dynamics of the environment and reward variations, while the linear nature of CVaR limits its adaptability in complex scenarios, leading to instability in training. Secondly, the CVaR distortion function only focuses on the tail regions of the distribution, ignoring other parts of the distribution. In reinforcement learning, rewards are typically a diverse signal containing various potential feedbacks from different states. By weighting only specific quantiles, CVaR may fail to fully utilize all available information, contributing to instability during training. Finally, the early stages of reinforcement learning are often accompanied by significant uncertainty and fluctuations, making it more difficult for the model to adapt to complex environments, which ultimately leads to training failure.

