# OpenReview forum: "HuRi : Humanoid Robots Adaptive Risk-ware Distributional Reinforcement Learning for Robust Control"
_ICLR.cc/2025/Conference — Submitted to ICLR 2025_

### Official Review · Reviewer_1gwC · 2024-11-01

**Soundness:** 3
**Presentation:** 1
**Contribution:** 3
**Rating:** 6
**Confidence:** 4

**Summary:**

The manuscript presents a novel method for risk-aware distributional reinforcement learning called HuRi. The main innovation compared to previous works is a novel adaptation scheme for choosing the risk-awareness sensitivity of the policy based on the current state (i.e. if the policy will be cautious, exploratory/aggressive or neutral). The authors propose to use a combination of distortion functions (already present in the literature) and the usage of a Random Network Distillation (RND) process. Overall, the new approach is evaluated in humanoid locomotion tasks and showcases superior performance compared to baselines and ablations.

**Strengths:**

1) Adapting risk-awareness depending on the current state is an interesting concept and has not been extensively explored. Especially in humanoid robots.

2) The usage of the RND process for risk adaptation is quite interesting and quite novel (as far as I am aware of the literature).

3) The real-world evaluation on a humanoid robot is highly appreciated.

4) The authors provide most details needed to replicate their approach.

**Weaknesses:**

1) The paper presentation needs a lot of work. There are numerous small things and typos that add up and make the paper difficult to understand. First, the citations are not well formatted and this makes reading the manuscript difficult. Secondly, the notation is not consistent: sometimes $\boldsymbol{x}$ is used for the state, and some other times $\boldsymbol{s}$ is used for the state. Another example is that the paper is about humanoid robots and then in the theorem the authors state "We describe the locomotion problem of quadruped robots". Overall, the presentation and writing of the paper requires quite some work.

2) As far as I can tell (given the difficulty of understanding the manuscript described above), the authors claim that the novelty lies in a) the usage of an adaptive mechanism for adapting the β parameter of the distortion function, and b) the training of the RND network for β adaptation. Although I can see value in the adaptation mechanism, the motivation of using the RND network is rather weak. Similarly, the evaluation and discussion on why it works better is again rather weak. I would have expected a more thorough discussion and analysis since RND seems to be the critical factor in the performance gains.

3) The experimental section requires more work. For example, in Fig. 3 (and the describing text) we do not know if the authors ran just one training and smoothed out rewards or they ran multiple seeds and take some statistics. Similarly, for the experiments shown in Fig. 4, the authors do not mention how many seeds they used. They do mention the number of seeds for the experiments shown in Tab. 1. Overall, this goes back to presentation; the manuscript requires substantial work to make it understandable.

4) In the current form, we cannot really evaluate the real-world experiments. Figures 6,7 and 8 do not showcase anything that is written in the text. We cannot validate what's happening here. The video showcases a lot of nice behaviors, but we need a more rigorous evaluation here. It is very difficult to evaluate the performance of the policy at the moment.

5) I do not get how the proposed approach is specifically suited for humanoids (which the author attempt to motivate, but fail imho).

**Questions:**

See weaknesses.

---

> ### Author Response · Authors · 2024-11-23
> **Response to Reviewer 1gwC**
>
> We sincerely appreciate your valuable comments and the time you have taken to review our paper. In the following sections, we address the concerns you raised and summarize the revisions made to the paper. To facilitate your review of these modifications, we have attached the revised manuscript.
> >   The paper presentation needs a lot of work. There are numerous small things and typos that add up and make the paper difficult to understand. First, the citations are not well formatted and this makes reading the manuscript difficult. Secondly, the notation is not consistent: sometimes x is used for the state, and some other times s is used for the state. Another example is that the paper is about humanoid robots and then in the theorem the authors state "We describe the locomotion problem of quadruped robots". Overall, the presentation and writing of the paper requires quite some work.
>
> A: In the revised version, we corrected all citation formatting to improve clarity. We also unified the notation throughout the manuscript to avoid ambiguity. Furthermore, we extensively revised the paper to eliminate typos and optimized its structure to enhance readability. Thank you very much for your careful review.
>
> > As far as I can tell (given the difficulty of understanding the manuscript described above), the authors claim that the novelty lies in a) the usage of an adaptive mechanism for adapting the β parameter of the distortion function, and b) the training of the RND network for β adaptation. Although I can see value in the adaptation mechanism, the motivation of using the RND network is rather weak. Similarly, the evaluation and discussion on why it works better is again rather weak. I would have expected a more thorough discussion and analysis since RND seems to be the critical factor in the performance gains.
>
> A: In the revised version, we clarified the motivation for using RND in **lines 300-303**. Previous studies have categorized uncertainty into intrinsic uncertainty and parameter uncertainty. To more accurately measure environmental uncertainty, we used the IQR module to measure intrinsic uncertainty and the RND module to measure parameter uncertainty. To further validate the roles of RND and IQR, we conducted ablation studies detailed in **Appendix A.5**. The results show that combining both types of uncertainty yields better performance than measuring a single type of uncertainty.
>
> > The experimental section requires more work. For example, in Fig. 3 (and the describing text) we do not know if the authors ran just one training and smoothed out rewards or they ran multiple seeds and take some statistics. Similarly, for the experiments shown in Fig. 4, the authors do not mention how many seeds they used. They do mention the number of seeds for the experiments shown in Tab. 1. Overall, this goes back to presentation; the manuscript requires substantial work to make it understandable.
>
> A: In the revised version, we significantly improved the descriptions of the experimental settings. We randomly selected five seeds for each method and repeated the training process five times. The aggregated results are presented in the updated **Figure 5**, further validating the stability and feasibility of our method. Additionally, we explicitly stated the number of seeds and repetitions used for each experiment. Finally, we included hyperparameters and details of the training and testing processes in **Appendix A.3** to facilitate reader understanding.
>
> > In the current form, we cannot really evaluate the real-world experiments. Figures 6,7 and 8 do not showcase anything that is written in the text. We cannot validate what's happening here. The video showcases a lot of nice behaviors, but we need a more rigorous evaluation here. It is very difficult to evaluate the performance of the policy at the moment.
>
> A: Currently, in the revised version, we added an additional real-world experiment, demonstrating stable walking on four different ground friction conditions. Additionally, we are conducting comparisons between HuRi and the baselines on the physical robot. However, the testing process is time-consuming. We are working on obtaining quantitative experimental results for HuRi and the baselines, and we will update these results in the paper within the next two days. We greatly appreciate your patience and understanding.

---

> ### Author Response · Authors · 2024-11-23
> **Response to Reviewer 1gwC**
>
> Connect to the previous response:
>
> > I do not get how the proposed approach is specifically suited for humanoids (which the author attempt to motivate, but fail imho).
>
> A: Since humanoid robots rely on bipedal balance control, they are more prone to losing balance under external disturbances compared to other robots. Our approach mitigates this issue by perceiving risks and adjusting the risk preference of the policy accordingly. We demonstrated in both simulation and real-world experiments that our method outperforms others in terms of stability and robustness. While we argue that our method is particularly suitable for humanoid robots, it is not limited to them. Additional experiments in **Appendix A.6** show that our method is also effective on quadruped robots.

---

> > ### Comment · Reviewer_1gwC · 2024-11-25
> > **Reply to rebuttal**
> >
> > >  In the revised version, we corrected all citation formatting to improve clarity
> >
> > The paper is in much better state now. Thank you.
> >
> > > The results show that combining both types of uncertainty yields better performance than measuring a single type of uncertainty.
> >
> > I appreciate the new results and explanation. My comment now becomes "the performance gain is not significant enough" to justify a whole novel approach. Overall, it's hard to see the algorithmic contribution of the manuscript.
> >
> > > In the revised version, we significantly improved the descriptions of the experimental settings.
> >
> > I appreciate the improvement. This now reads much better and makes the results more reliable/convincing.
> >
> > > Currently, in the revised version, we added an additional real-world experiment, demonstrating stable walking on four different ground friction conditions. Additionally, we are conducting comparisons between HuRi and the baselines on the physical robot.
> >
> > I appreciate the effort and the new experiments.
> >
> > > While we argue that our method is particularly suitable for humanoid robots, it is not limited to them.
> >
> > Then why argue that your method is particularly suitable for humanoid robots?
> >
> > **Verdict**
> >
> > Overall, I appreciate the efforts by the authors, and I will slightly increase my score. But I would really like to see the quantitative results on the physical robot.

---

> > > ### Author Response · Authors · 2024-12-02
> > > **Response to Reviewer 1gwC**
> > >
> > > Dear Reviewer 1gwC,
> > >
> > > First and foremost, we sincerely thank you for your valuable feedback on our paper. Based on your suggestions, we have made several revisions, which are summarized as follows:
> > >
> > > ● We have added additional experimental comparisons with baselines in real-world experiments.
> > >
> > > ● We have further elaborated on why our approach is suitable for humanoid robots.
> > >
> > > Once again, thank you for your time and effort in reviewing our work. We look forward to your feedback.
> > >
> > > Best regards,
> > >
> > > Authors of 14098

---

> > > > ### Comment · Reviewer_1gwC · 2024-12-03
> > > >
> > > > I am satisfied by the authors' replies and new experiments. I highly appreciate the effort. I do believe that this is an interesting paper with nice results, but I still believe that the algorithmic novelty is minor. I would consider raising my recommendation by one point (6->7), but the system does not allow this, and I believe that giving a grade of 8 is too much for the considered algorithmic novelty. With this reasoning, I am keeping my score, but tending towards increasing it.

---

> > > > > ### Author Response · Authors · 2024-12-03
> > > > > **Response to Reviewer 1gwC**
> > > > >
> > > > > Thank you for your detailed review of our paper and for your high appreciation of our new experiments and efforts. We are grateful for your recognition of the results presented in the paper and understand your evaluation regarding the novelty of the algorithm.
> > > > >
> > > > > Regarding the algorithmic novelty, we would like to emphasize that our work is the first to comprehensively consider both intrinsic and parameter uncertainty in the environment. It also represents the first attempt to combine the RND and IQR modules, explicitly incorporating uncertainty evaluation into the algorithm. This approach promotes safe exploration by the agent during training, thereby improving the generalization ability of the model. Extensive experiments have demonstrated the effectiveness of our method on robots. Additionally, we have validated the contribution of each module through ablation studies. We will further highlight the differences between our algorithm and existing methods in the paper and explore more breakthrough algorithmic improvements in future work.
> > > > >
> > > > > We will also include more discussions in the paper to ensure that the novelty of the algorithm is clearly conveyed. We sincerely appreciate your consideration of our score being above 6. We noticed that you found value in certain aspects of the paper and were inclined to give a score of 7, but due to the absence of a 7-point option in the system, you selected a score of 6. We fully understand that a score of 6 reflects your positive evaluation of the paper, and we greatly appreciate your feedback. Based on this, we would like to emphasize that, considering the contributions of our work in algorithmic novelty, experimental validation, and application scenarios, a score of 8 would better align with the efforts and results we have achieved in these areas. We believe that further improvements and future work will continue to demonstrate the potential of our method.
> > > > >
> > > > > Once again, thank you for your valuable time and feedback. Regardless of the final score, we will carefully consider your suggestions and continue to refine and improve our research.

---

> ### Comment · Area_Chair_oWim · 2024-11-25
>
> Dear Reviewer,
>
> Please provide feedback to the authors before the end of the discussion period, and in case of additional concerns, give them a chance to respond.
>
> Timeline: As a reminder, the review timeline is as follows:
>
> November 26: Last day for reviewers to ask questions to authors.
>
> November 27: Last day for authors to respond to reviewers.

---

> ### Author Response · Authors · 2024-11-27
> **Response to Reviewer 1gwC**
>
> Thank you for recognizing our efforts in revising the paper. We sincerely appreciate your valuable comments and the time you have taken to review our paper. We address the concerns you raised:
> > My comment now becomes "the performance gain is not significant enough" to justify a whole novel approach. Overall, it's hard to see the algorithmic contribution of the manuscript.
>
> A: In terms of the algorithm, we propose for the first time a comprehensive approach to measure uncertainty in the environment by considering both intrinsic uncertainty and parameter uncertainty. By combining these two types of uncertainty, we can more accurately assess the environmental uncertainty. To demonstrate the superiority of our method, we tested it under out-of-distribution disturbances. In the presence of external force interference, our method improved the success rate by nearly 20% compared to the baseline. Under load and mass disturbances, the velocity tracking error of our method consistently remained the lowest. This advantage is even more pronounced in real-world experiments. In scenarios involving external forces on the robot, the success rate of our method (65%) was significantly higher than that of the baseline (35%). When additional loads were applied to the center of mass and feet, we tested the velocity error rates at speeds of 0.3 m/s, 0.6 m/s, and 0.9 m/s. Our method consistently maintained the lowest velocity error rate. With the inclusion of quantitative real-world experiments, the advantages of our method became even more evident.
> > Then why argue that your method is particularly suitable for humanoid robots?
>
> A: We would like to provide further clarification on this point. Compared to quadruped robots, wheeled robots, and humanoid robots exhibit smaller stability margins, making them more sensitive to disturbances. Our method, through distributional reinforcement learning, allows for risk modeling of the surrounding environment (such as terrain noise, sensor noise, etc.) during the robot's walking process. This guides the robot to explore the action space more safely and efficiently, leading to improved generalization and robustness in real-world deployments. While previous research on safety reinforcement learning has achieved success on quadruped robots, quadrupeds have larger stability margins, and recent studies have demonstrated that they are capable of overcoming, rather than avoiding, such risks. In contrast, for the current and future application scenarios of humanoid robots, robustness and safe walking are more critical than agility. Therefore, we place greater emphasis on risk-aware policies, which makes our approach particularly well-suited for humanoid robots.
>
> > I would really like to see the quantitative results on the physical robot.
>
> A: We have updated the revised version. In the revised version, we included quantitative real-world experiments to further validate the effectiveness of our approach. First, we employed a pendulum system to generate a fixed lateral impact force and tested the success rate of the robot at a velocity of 0.6 m/s. A failure was defined as the robot falling over. Experimental results demonstrated that our method achieved the highest success rate, outperforming other methods. Additionally, we evaluated the velocity error rate under the condition of adding a 5 kg load to the robot's center of mass. Through these experiments, we further assessed the impact of load on the robot's stability. Another setting of experiments was designed to apply tensile force on the robot's legs by adding loads to its feet, testing the velocity tracking error rate under this condition. The results showed that, in both scenarios, our method consistently maintained the lowest velocity error rate, demonstrating that our method is more effective at resisting out-of-distribution disturbances. These experiments verified the effectiveness of our method under complex conditions. We firmly believe that these results clearly highlight the superiority of our approach, fully addressing the reviewers' concerns.
>
> Thank you for your detailed review and valuable suggestions. We hope that our revisions have addressed all your concerns and questions. Please let us know if there are any remaining issues.

---

> ### Author Response · Authors · 2024-11-29
> **Response to Reviewer 1gwC**
>
> Dear Reviewer 1gwC,
>
> ﻿
> We sincerely appreciate your insightful comments and valuable suggestions on improving our work. As the final deadline approaches, we would kindly like to ask whether our responses have sufficiently addressed your questions and concerns. We are happy to engage in further discussion or provide any clarifications needed, and we welcome any additional feedback to strengthen our work before the rebuttal period ends.
> ﻿
> Thank you for your time and thoughtful reviews.
> ﻿
>
> Kind regards,
>
> ﻿
> Authors

---

### Official Review · Reviewer_7TAc · 2024-11-04

**Soundness:** 2
**Presentation:** 1
**Contribution:** 2
**Rating:** 5
**Confidence:** 4

**Summary:**

## Humanoid Robotcs Adaptive Risk-aware Distributional RL for Robust Control
This paper presents a risk-aware framework for training humanoid robot policies. The main contribution of the paper is the usage of distributional RL as a method for controlling the risk that a humanoid's policy incurs during exploration and execution. The motivation of the risk-awareness in humanoid systems comes from the delicate and robust control that the actuators must execute on the robot in order to maintain stability. Finding these stable controllers is difficult and dangerous, so risk-awareness in the policy is necessary. The authors provide experiments on a simulated humanoid robot and real-world experiments.

**Strengths:**

This paper is well-motivated and attempts to solve a real-world problem in humanoid robotics. The demonstration of their method on real robots is commendable and a good contribution to the paper.

**Weaknesses:**

There are a series of minor typo errors as well as incorrect latex formatting. While most of these issues don't hinder understanding, they make it harder to read the paper. For example:
1. Title says "Risk-ware" instead of "risk-aware"
2. Line 106 is missing a citation
3. Most citations seem to not use \citep{} or \citet{}. They all are joined with the previous word.

Major issues:
1. Presentation of method is unclear: throughout section 3, the authors introduce various components of their method. Because each component is described mainly in text, the structure of dependancies and framework is difficult to understand. Figure 2 is helpful, but not fully informative. I would suggest including an Algorithm Block that goes through each component step-by-step. Further, it would be helpful to include labels for all the equations/expressions that are splitting up the paragraphs throughout the method section
2. Usage of $SR(\lambda)$: This term is introduced around line 229 but never really described in great detail. In general, this paragraph from 226 to 240 is confusing and would benefit from being described through an algorithm block
3. The quantile loss in line 241 does not include the distribution over which the expectations are being taken.
4. MSE loss in line 247 has incorrect notation that make its meaning ambiguous (mismatched paranthesis)
5. $g_\beta^\text{Wang}$ in line 254 is introduced without any additional supporting information. In the following paragraph, the authors describe the nuances of $\beta$ but not where the function $g$ is ever used. Please clarify its position in the method.
6. *Poor reward comparison experiment*: The results in Figure 3 do not include any error bars or indicate any notion of statistical significance. It is necessary for the authors to run these experiments over multiple seeds and report aggregated performance. Also, is reward the right metric of performance? I would imagine that violation of safety constraints is a more important feature to report on.
7. Robot specific simulated experiments: While the results reported in the remainder of the simulated experiment section are important, they seem more like results that would be found in a pure robotics paper.
8. Real world experiments lack statistical or more analytical significance. Please include some information on how the experiments show that the risk-awareness is important and that your instantiation of the risk-aware framework out performs other real-world methods. To do this, you might report success rates and safety violations that the varying methods exhibit.

**Questions:**

No additional questions. Please address the issues/questions I mention in the Weakness section

---

> ### Author Response · Authors · 2024-11-23
> **Response to Reviewer 7TAc**
>
> We are grateful for your valuable comments. In the subsequent sections, we address the concerns you raised. For your convenience in reviewing these modifications, we have attached the revised manuscript.
> > There are a series of minor typo errors as well as incorrect latex formatting. While most of these issues don't hinder understanding, they make it harder to read the paper.
>
> A: In the revised version, we have corrected these spelling and formatting issues and improved the organization and language of the paper. Thank you very much for your careful review.
>
> > Presentation of method is unclear: throughout section 3, the authors introduce various components of their method. Because each component is described mainly in text, the structure of dependancies and framework is difficult to understand. Figure 2 is helpful, but not fully informative. I would suggest including an Algorithm Block that goes through each component step-by-step. Further, it would be helpful to include labels for all the equations/expressions that are splitting up the paragraphs throughout the method section.
>
> A: In the revised version, we improved the organization of the method section by assigning sub-section titles to each module, making the structure clearer. We also updated **Figure 2** to better illustrate the relationships and dependencies among components. Additionally, we included a detailed algorithm block in **Appendix A.4** to describe the full step-by-step process.
>
> > Usage of SR(λ): This term is introduced around line 229 but never really described in great detail. In general, this paragraph from 226 to 240 is confusing and would benefit from being described through an algorithm block.
>
> A: To provide a clearer explanation of the SR(λ) method, we added a detailed description of its algorithmic process in **Appendix A.4** of the revised version.
>
> > The quantile loss in line 241 does not include the distribution over which the expectations are being taken.
>
> A: In the revised version, we clarified that $\mathcal{T} Z_\theta(s)$ is the target distribution in **line 236**. The energy loss calculated in the equation represents the difference between the Critic's output distribution and the target distribution.
>
> > MSE loss in line 247 has incorrect notation that make its meaning ambiguous (mismatched paranthesis).
>
> A: In the revised version, we corrected this notation error. Thank you for pointing this out and for your careful review.
>
> > $g_\beta^\text{Wang}(\tau)$ in line 254 is introduced without any additional supporting information. In the following paragraph, the authors describe the nuances of β but not where the function g is ever used. Please clarify its position in the method.
>
> A: In the revised version, we clarified in line 264 that  represents $h_\beta^\text{Wang}(\tau)$ the distorted quantile. To avoid confusion with the target network symbol 𝑔 introduced later, we replaced the notation $g_\beta$ with $h_\beta$.
>
> > Poor reward comparison experiment: The results in Figure 3 do not include any error bars or indicate any notion of statistical significance. It is necessary for the authors to run these experiments over multiple seeds and report aggregated performance. Also, is reward the right metric of performance? I would imagine that violation of safety constraints is a more important feature to report on.
>
> A: In the revised version, we selected five random seeds for each method and repeated the training process five times. The aggregated results are presented in the updated **Figure 3**. These results further demonstrate the stability and feasibility of our method. Additionally, in **lines 72-79**, we explain that our method promotes safe exploration during training compared to other approaches that use fixed scalar risk parameters. To support this claim, we presented results showing higher rewards achieved by our method, which reflect the effectiveness of safe exploration during training.
>
> > Robot specific simulated experiments: While the results reported in the remainder of the simulated experiment section are important, they seem more like results that would be found in a pure robotics paper.
>
> A: In the revised version, beyond the simulation experiment section, we added several extended experiments in Appendix. These include: Ablation studies demonstrating the effectiveness of jointly estimating intrinsic and parameter uncertainties; Experiments on a quadruped robot to verify that our method does not depend on specific reward structures; Multi-terrain testing that confirms our method improves the robot's motion control capabilities; Comparisons with the CVaR distortion function to show that the Wang function is more compatible with our approach. These additional experiments further highlight the generalizability and theoretical soundness of our method.

---

> ### Author Response · Authors · 2024-11-23
> **Response to Reviewer 7TAc**
>
> Connect to the previous response:
> > Real world experiments lack statistical or more analytical significance. Please include some information on how the experiments show that the risk-awareness is important and that your instantiation of the risk-aware framework out performs other real-world methods. To do this, you might report success rates and safety violations that the varying methods exhibit.
>
> A: Currently, in the revised version, we added an additional real-world experiment, demonstrating stable walking on four different ground friction conditions. Additionally, we are conducting comparisons between HuRi and the baselines on the physical robot. However, the testing process is time-consuming. We are working on obtaining quantitative experimental results for HuRi and the baselines, and we will update these results in the paper within the next two days. We greatly appreciate your patience and understanding.

---

> ### Comment · Area_Chair_oWim · 2024-11-25
>
> Dear Reviewer,
>
> Please provide feedback to the authors before the end of the discussion period, and in case of additional concerns, give them a chance to respond.
>
> Timeline: As a reminder, the review timeline is as follows:
>
> November 26: Last day for reviewers to ask questions to authors.
>
> November 27: Last day for authors to respond to reviewers.

---

> ### Comment · Reviewer_7TAc · 2024-11-27
> **Thank you for revisions**
>
> The clarity of the updated manuscript has improved since the review process began. However, the manuscript is still lacking the rigorous, detailed, and most importantly, clear, descriptions of the method and reasoning. For instance, the expectations in equations 3,4, and 5 still do not have the distributions over which they are taken.

---

> > ### Author Response · Authors · 2024-11-27
> > **Response to Reviewer 7TAc**
> >
> > We sincerely appreciate your recognition of our efforts in revising the paper. Thank you for your valuable comments and the time you dedicated to reviewing our work. We have addressed the concerns you raised as follows:
> > > However, the manuscript is still lacking the rigorous, detailed, and most importantly, clear, descriptions of the method and reasoning. For instance, the expectations in equations 3,4, and 5 still do not have the distributions over which they are taken.
> >
> > A: In the revised version, we have added descriptions of the variables for Equations 3, 4, and 5. We understand your point. The expectations calculated in the equations are based on random variables that are outputs of the model. In **lines 214-217**, we define the relevant random variables. These random variables correspond to quantiles predicted by the neural network, which represent the action-value distribution. These quantiles do not follow a fixed distribution. Therefore, the expectations are computed based on the distribution learned by the model, rather than a predefined fixed distribution.  Thank you again for your valuable feedback.
> >
> > > Real world experiments lack statistical or more analytical significance.
> >
> > A: We have updated the revised version. In the revised version, we included quantitative real-world experiments to further validate the effectiveness of our approach. First, we employed a pendulum system to generate a fixed lateral impact force and tested the success rate of the robot at a velocity of 0.6 m/s. A failure was defined as the robot falling over. Experimental results demonstrated that our method achieved the highest success rate, outperforming other methods. Additionally, we evaluated the velocity error rate under the condition of adding a 5 kg load to the robot's center of mass. Through these experiments, we further assessed the impact of load on the robot's stability. Another setting of experiments was designed to apply tensile force on the robot's legs by adding loads to its feet, testing the velocity tracking error rate under this condition. The results showed that, in both scenarios, our method consistently maintained the lowest velocity error rate, demonstrating that our method is more effective at resisting out-of-distribution disturbances. These experiments verified the effectiveness of our method under complex conditions. We firmly believe that these results clearly highlight the superiority of our approach, fully addressing the reviewers' concerns.
> >
> > We sincerely appreciate your valuable comments once again. If there are any errors in the theoretical description, we kindly ask you to point out the specific issues in the derivations or any errors in the notation. We highly value the theoretical part, but we have also clearly outlined the algorithm's process in the **Appendix A.4**. Furthermore, with the theoretical logic properly clarified, our experimental results have demonstrated significant advancements, particularly in enhancing the generalization of humanoid robot locomotion tasks. We hope the reviewer can provide a more balanced evaluation, considering both the theoretical and experimental aspects, rather than relying solely on the theoretical part as the sole criterion for assessment.

---

> > ### Author Response · Authors · 2024-11-29
> > **Response to Reviewer 7TAc**
> >
> > Dear Reviewer 7TAc,
> >
> >
> > ﻿ We sincerely appreciate your insightful comments and valuable suggestions on improving our work. As the final deadline approaches, we would kindly like to ask whether our responses have sufficiently addressed your questions and concerns. We are happy to engage in further discussion or provide any clarifications needed, and we welcome any additional feedback to strengthen our work before the rebuttal period ends. ﻿ Thank you for your time and thoughtful reviews. ﻿
> >
> >
> > Kind regards,
> >
> >
> > ﻿ Authors

---

> > ### Author Response · Authors · 2024-12-03
> > **Response to Reviewer 7TAc**
> >
> > Dear Reviewer 7TAc,
> >
> > ﻿ We sincerely appreciate your insightful comments and valuable suggestions on improving our work. As the final deadline approaches, we would kindly like to ask whether our responses have sufficiently addressed your questions and concerns. We are happy to engage in further discussion or provide any clarifications needed, and we welcome any additional feedback to strengthen our work before the rebuttal period ends.﻿ Please kindly let us know if you have additional questions or concerns that stand between us and a higher score! Thank you for your time and thoughtful reviews. ﻿
> >
> > Kind regards,
> >
> > ﻿ Authors

---

> ### Author Response · Authors · 2024-12-02
> **Response to Reviewer 7TAc**
>
> Dear Reviewer 7TAc,
>
> First and foremost, we sincerely thank you for your valuable feedback on our paper. Based on your suggestions, we have made several revisions, which are summarized as follows:
>
> ● We have provided clarifications for certain terms and experimental details, and have modified sections that could lead to misunderstandings.
>
> ● We have added additional experimental comparisons with baselines in real-world experiments.
>
> ● We have included additional ablation studies in the appendix to further substantiate our method.
>
> Once again, thank you for your time and effort in reviewing our work. We look forward to your feedback.
>
> Best regards,
>
> Authors of 14098

---

> ### Comment · Reviewer_7TAc · 2024-12-03
> **Thank you for the revisions**
>
> Hello Authors,
>
> The clarifications you have made do in fact help the paper and I appreciate the effort. The increased clarity helps the paper and I am willing to increase my score to a 5.
>
> However, I am still not satisfied with the presentation of the technical details and preliminaries, especially in sections 3.2 and 3.3. There is still a lack of rigor in the presentation of the mathematical elements of the method.
>
> I appreciate all your efforts in improving the paper thus far.
>
> Best,
> Reviewer 7TAc

---

> > ### Author Response · Authors · 2024-12-03
> > **Response to Reviewer 7TAc**
> >
> > Dear Reviewer 7TAc,
> >
> > Thank you very much for your thoughtful and constructive feedback. We truly appreciate your recognition of the improvements we have made. We are glad that the clarifications have helped enhance the clarity of the paper.
> >
> > We understand the importance of rigor in presenting the mathematical elements of the method, and we sincerely appreciate your valuable input on this matter. To address your concern, we will provide a detailed explanation of the meaning of each mathematical symbol in Appendix and rigorously derive the update rules for the distributional reinforcement learning method.
> >
> > Once again, thank you for your time and thoughtful comments. We will carefully incorporate your suggestions to further improve the quality of the paper.
> >
> > Best regards,
> > Authors of 14098

---

### Official Review · Reviewer_2vVJ · 2024-11-04

**Soundness:** 3
**Presentation:** 3
**Contribution:** 3
**Rating:** 8
**Confidence:** 4

**Summary:**

This paper introduces HuRi, a new method for training humanoid robots to walk stably in challenging conditions by automatically adjusting their movement cautiousness based on environmental risks. The approach combines distributional reinforcement learning, which learns the full range of possible outcomes rather than just averages, with an adaptive risk adjustment system that uses uncertainty measures (IQR and RND) to assess danger levels and modify the robot's behavior accordingly. The authors demonstrate through both simulated and real-world experiments that HuRi outperforms baseline methods, achieving better stability under various disturbances including external forces, heavy loads (up to 42% of body weight), and sudden impacts.

**Strengths:**

- Well written paper with thorough implementation details. The empirical results demonstrates meaningful improvements over baselines for humanoid walking, particularly in handling disturbances and varied terrain

- Effectively demonstrates how uncertainty-aware RL can aid safe exploration in complex humanoid tasks - an important contribution that could extend beyond locomotion to manipulation and other high-dimensional control challenges in humanoid robotics

**Weaknesses:**

- Statistical robustness concerns: Figure 3 appears to show results from a single training seed, as evidenced by sharp performance dips. Multiple seeds are needed to validate learning stability, and success rate curves across seeds would better demonstrate robustness claims

- Limited exploration of reward structure impacts: While the method shows improvements, there's no analysis of how dependent these gains are on their specific reward formulation. Recent work [1] has shown phase/clock-based rewards may not be optimal for robust locomotion. A comparison with alternate reward structures would strengthen their claims.

- Uncertainty measurement complexity: The combination of RND with distributional RL seems potentially involved. I did not quite understand the justification of why RND is needed on top of distributional uncertainty estimates. Did the authors try to explore whether simpler approaches could achieve similar results, for example just using RND?

**Questions:**

**Minor Details:**

1. For improved readability, best scores in Table 1 should be highlighted in bold or black

2. Section 4.2's experimental description needs clearer organization: explicitly state that the evaluation focuses on two distinct robustness tasks (handling continuous forces and sudden impacts) with corresponding training setups. Current wording creates ambiguity about whether evaluations were conducted on out-of-distribution scenarios



*[1] Marum, Bart Jaap van et al. “Revisiting Reward Design and Evaluation for Robust Humanoid Standing and Walking.” *ArXiv* abs/2404.19173 (2024): n. pag.*

---

> ### Author Response · Authors · 2024-11-23
> **Response to Reviewer 2vVJ**
>
> Thank you for taking the time to review our paper and for providing such helpful comments! We are glad that you like our ideas. You have raised several valuable suggestions, which we address below. Please let us know if you have any remaining questions or concerns.
> >Statistical robustness concerns: Figure 3 appears to show results from a single training seed, as evidenced by sharp performance dips. Multiple seeds are needed to validate learning stability, and success rate curves across seeds would better demonstrate robustness claims.
>
> A: In the revised version, we selected five random seeds for each method and repeated the training process five times. The experimental results are shown in the updated **Figure 3**, further validating the stability and feasibility of our method.
>
> >Limited exploration of reward structure impacts: While the method shows improvements, there's no analysis of how dependent these gains are on their specific reward formulation. Recent work [1] has shown phase/clock-based rewards may not be optimal for robust locomotion. A comparison with alternate reward structures would strengthen their claims.
>
> A: To demonstrate that our method does not depend on specific reward formulations, we trained and tested our approach on a quadruped robot using other reward formulations. Details of the training and testing are provided in **Appendix A.6**. The training results show that our method achieved higher rewards and reached higher terrain levels faster than the baselines, further proving its ability for safe exploration during training. During testing, we assigned the quadruped robot a challenging and risky task—walking down from a high platform. Our method achieved a significantly higher survival rate compared to the baseline. These experiments demonstrate that our method performs well under different reward formulations.
>
> >Uncertainty measurement complexity: The combination of RND with distributional RL seems potentially involved. I did not quite understand the justification of why RND is needed on top of distributional uncertainty estimates. Did the authors try to explore whether simpler approaches could achieve similar results, for example just using RND?
>
> A: In the revised version, we clarified in **lines 282–289** of the manuscript that environmental uncertainty is divided into intrinsic uncertainty and parameter uncertainty. Intrinsic uncertainty can be directly measured using the IQR module for return distributions. To estimate environmental uncertainty more comprehensively, we used the RND module. The motivation for the RND module is explained in **lines 300-303**. We also conducted ablation studies, the details of which are presented in **Appendix A.5**. In these experiments, we tested a method that uses only the RND module. The results show that our method outperforms the approach with only the RND module, demonstrating that combining both types of uncertainties is more effective. Detailed analyses are provided in **Appendix A.5**.
>
> >For improved readability, best scores in Table 1 should be highlighted in bold or black.
>
> A: In the revised version, the best scores in **Table 1** are highlighted in bold. Thank you very much for your thorough and thoughtful review.
>
> > Section 4.2's experimental description needs clearer organization: explicitly state that the evaluation focuses on two distinct robustness tasks (handling continuous forces and sudden impacts) with corresponding training setups. Current wording creates ambiguity about whether evaluations were conducted on out-of-distribution scenarios.
>
> A: In the revised version, we clarified the experimental settings to ensure that readers understand our evaluations were conducted on out-of-distribution scenarios. Additionally, we included a table in **Appendix A.3** listing the domain randomization parameters used during training and testing to emphasize that our evaluation was conducted under out-of-distribution disturbances.
>
> [1]Marum, Bart Jaap van et al. “Revisiting Reward Design and Evaluation for Robust Humanoid Standing and Walking.” ArXiv abs/2404.19173 (2024): n. pag.

---

> ### Comment · Area_Chair_oWim · 2024-11-25
>
> Dear Reviewer,
>
> Please provide feedback to the authors before the end of the discussion period, and in case of additional concerns, give them a chance to respond.
>
> Timeline: As a reminder, the review timeline is as follows:
>
> November 26: Last day for reviewers to ask questions to authors.
>
> November 27: Last day for authors to respond to reviewers.

---

### Official Review · Reviewer_18Yh · 2024-11-04

**Soundness:** 3
**Presentation:** 2
**Contribution:** 2
**Rating:** 5
**Confidence:** 5

**Summary:**

This paper presents an approach to training humanoid robots for locomotion in risky environments using deep reinforcement learning.  The key innovation is the development of an adaptive risk-aware distributional reinforcement learning algorithm, which allows adjusting risk sensitivity based on uncertainty in the environment.  This is achieved by combining intrinsic uncertainty, measured using the interquartile range, and parameter uncertainty, evaluated using random network distillation.  The proposed algorithm enables the robot to learn a robust locomotion control policy.  The method's effectiveness is demonstrated through simulations and real-world experiments on the Zerith robot, showing its ability to handle various disturbances such as sudden impacts, load variations, and uneven terrain

**Strengths:**

1. This paper introduce Wang function for risk-aware reinforcement learning and considers intrinsic uncertainty and parameter uncertainty for risk sensitivity adjustment.
2. The proposed method demonstrates superior performance compared to the baseline PPO and a fixed risk-sensitive method (CVaR0.5) in both simulation.
3. The proposed method is extensively evaluated in different scenarios, including simulations with different types of disturbances and real-world experiments on a physical robot.

**Weaknesses:**

1. Effectiveness of Wang function needs to be verified. Considering add some experiments to verify the effectiveness of Wang function as it is one of the core contribution of this paper, for example, adjusting the risk sensitivity of CvaR with intrinsic uncertainty and parameter uncertainty and compare with your method.
2. The real robot experiment is weak, authors should considering comparing your method and baselines on real robot.
3. The authors claim that "our method exhibits strong robustness in agents and bridges the gap between simulation environments and the real world", however, there is no content in the paper shows how the method benefits sim2real.
3. Considering other recent work focusing on blind humanoid locomotion, such as [1], have much better performance on locomotion, I wonder if risk-aware reinforcement learning is harmful to locomotion task.

[1] Advancing humanoid locomotion: Mastering challenging terrains with denoising world model learning, Xinyang Gu, Yen-Jen Wang, Xiang Zhu, Chengming Shi, Yanjiang Guo, Yichen Liu, Jianyu Chen, RSS 2024

**Questions:**

See weakness.

---

> ### Author Response · Authors · 2024-11-23
> **Response to Reviewer 18Yh**
>
> We sincerely appreciate your valuable feedback and the time you have taken to review our paper. In the following, we address the issues you raised and summarize the revisions made to the paper.
> >Effectiveness of Wang function needs to be verified. Considering add some experiments to verify the effectiveness of Wang function as it is one of the core contribution of this paper, for example, adjusting the risk sensitivity of CVaR with intrinsic uncertainty and parameter uncertainty and compare with your method.
>
> A: In the revised version, we employed the CVaR distortion function in combination with the RND module and the IQR module to adaptively adjust the risk sensitivity. This experiment is documented in **Appendix A.8**. The results indicate that, despite trying multiple hyperparameter configurations, using the CVaR distortion function consistently led to training failures. We have conducted a detailed analysis of the reasons for these failures in **Appendix A.8**, further demonstrating that the Wang function is more compatible with the RND and IQR modules.
>
> >The real robot experiment is weak, authors should considering comparing your method and baselines on real robot.
>
> A: Currently, in the revised version, we added an additional real-world experiment, demonstrating stable walking on four different ground friction conditions. Additionally, we are conducting comparisons between HuRi and the baselines on the physical robot. However, the testing process is time-consuming. We are working on obtaining quantitative experimental results for HuRi and the baselines, and we will update these results in the paper within the next two days. We greatly appreciate your patience and understanding.
>
> >The authors claim that "our method exhibits strong robustness in agents and bridges the gap between simulation environments and the real world", however, there is no content in the paper shows how the method benefits sim2real.
>
> A: We sincerely apologize for the inaccurate wording in the original manuscript. In the revised version, we have updated the statement to read: "we demonstrate that our method exhibits strong robustness in agents and successfully validates sim-to-real transfer." Additionally, we have detailed our sim-to-real settings in **lines 480-483**.
>
> >Considering other recent work focusing on blind humanoid locomotion, such as [1], have much better performance on locomotion, I wonder if risk-aware reinforcement learning is harmful to locomotion task.
>
> A: Current humanoid robots navigating extremely complex terrains, such as stairs, often require additional historical information inputs. Partial observations can only capture local information and fail to comprehensively reflect the complexity of the environment. Many existing methods, such as [1], address partial observability by incorporating historical information to expand the observation space. Our method focuses on enhancing safe exploration during the training process to improve the model's generalization ability, enabling it to resist out-of-distribution disturbances. Therefore, it does not incorporate historical information as input. We further validated that our method does not negatively impact locomotion tasks. First, we trained our method and baselines on multi-terrain environments and tested these models. The experimental settings and results are presented in **Appendix A.7**. The results show that our method achieved the highest reward and terrain level during training and the highest survival rate under external force disturbances during testing. Second, we conducted additional multi-terrain tests on a quadruped robot. The training setup and testing results are also included in **Appendix A.6**. During testing, we assigned the quadruped robot a challenging and risky task—walking down from a high platform. Our method achieved a significantly higher survival rate compared to the baseline. Both experiments conducted on multi-terrain settings demonstrate that our method does not harm locomotion tasks.
>
> [1] Advancing humanoid locomotion: Mastering challenging terrains with denoising world model learning, Xinyang Gu, Yen-Jen Wang, Xiang Zhu, Chengming Shi, Yanjiang Guo, Yichen Liu, Jianyu Chen, RSS 2024

---

> > ### Author Response · Authors · 2024-11-29
> > **Response to Reviewer 18Yh**
> >
> > Dear Reviewer 18Yh,
> >
> >
> > ﻿ We sincerely appreciate your insightful comments and valuable suggestions on improving our work. As the final deadline approaches, we would kindly like to ask whether our responses have sufficiently addressed your questions and concerns. We are happy to engage in further discussion or provide any clarifications needed, and we welcome any additional feedback to strengthen our work before the rebuttal period ends. ﻿ Thank you for your time and thoughtful reviews. ﻿
> >
> >
> > Kind regards,
> >
> >
> > ﻿ Authors

---

> ### Comment · Area_Chair_oWim · 2024-11-25
>
> Dear Reviewer,
>
> Please provide feedback to the authors before the end of the discussion period, and in case of additional concerns, give them a chance to respond.
>
> Timeline: As a reminder, the review timeline is as follows:
>
> November 26: Last day for reviewers to ask questions to authors.
>
> November 27: Last day for authors to respond to reviewers.

---

> ### Author Response · Authors · 2024-11-27
> **Response to Reviewer 18Yh**
>
> Thank you for your patience. We have updated the revised version and incorporated your suggestions by adding comparisons with other baselines in the real-world experiments.
> >The real robot experiment is weak, authors should considering comparing your method and baselines on real robot.
>
> A: We have updated the revised version. In the revised version, we included quantitative real-world experiments to further validate the effectiveness of our approach. First, we employed a pendulum system to generate a fixed lateral impact force and tested the success rate of the robot at a velocity of 0.6 m/s. A failure was defined as the robot falling over. Experimental results demonstrated that our method achieved the highest success rate, outperforming other methods. Additionally, we evaluated the velocity error rate under the condition of adding a 5 kg load to the robot's center of mass. Through these experiments, we further assessed the impact of load on the robot's stability. Another setting of experiments was designed to apply force on the robot's legs by adding loads to its feet, testing the velocity tracking error rate under this condition. The results showed that, in both scenarios, our method consistently maintained the lowest velocity error rate, demonstrating that our method is more effective at resisting out-of-distribution disturbances. These experiments verified the effectiveness of our method under complex conditions. We firmly believe that these results clearly highlight the superiority of our approach, fully addressing the reviewers' concerns.
>
> Thank you for your thorough review and helpful suggestions. We hope we have addressed all your concerns and questions. Please let us know if there are any concerns.

---

> ### Author Response · Authors · 2024-12-02
> **Response to Reviewer 18Yh**
>
> Dear Reviewer 18Yh,
>
> First and foremost, we sincerely thank you for your valuable feedback on our paper. Based on your suggestions, we have made several revisions, which are summarized as follows:
>
> ● We have elaborated on and demonstrated the role of the Wang_function.
>
> ● We have provided clarifications for certain terms and experimental details, and have modified sections that could lead to misunderstandings.
>
> ● We have added additional experimental comparisons with baselines in real-world experiments.
>
> ● We have empirically demonstrated that our method does not compromise the robot's motion control capabilities.
>
> Once again, thank you for your time and effort in reviewing our work. We look forward to your feedback.
>
> Best regards,
>
> Authors of 14098

---

> > ### Comment · Reviewer_18Yh · 2024-12-02
> > **Response to Authors**
> >
> > Thanks the authors for providing experiments on Wang_function, which make the contribution of this paper more clear. I also thank the authors for the real robot experiments, although the proposed method outperforms baselines, it is not clear how the experiments are conducted, how do you measure the robot's velocity, how many times do you repeat the experiments, how the velocity error computed? I do not understand why observation history is not used, many previous work shows that observation history contributes to both the performance and sim2real. Considering these, I keep my current rate.

---

> > > ### Author Response · Authors · 2024-12-02
> > > **Response to Reviewer 18Yh**
> > >
> > > We sincerely appreciate your valuable feedback and the time you have taken to review our paper. You have raised several valuable suggestions, which we address below：
> > >
> > > > About the setup of the physical experiment
> > >
> > > Due to the limited length of the paper, we apologize for not providing a more detailed description of the experimental settings for the real-world experiments. The experimental settings for applying external forces to the robot is shown in the **Figure 6**. The settings for adding load to the robot's center of mass is also depicted in **Figure 7.B**. Additionally, the experimental configuration for adding load to the robot's feet is shown in **Figure 7.D**. We set fixed starting and ending points, with the robot walking 50 meters in each test. The time taken for the robot to walk is measured, from which we calculate the true velocity. For each experiment, we repeat the test twenty times and compute the average results. The velocity error rate is calculated using the following formula:
> > > \begin{equation}
> > > R_\text{error} = \left( \frac{| V_\text{true} - V_\text{command} |}{V_\text{true}} \right)
> > > \end{equation}
> > > Here, $R_\text{error}$ represents the velocity error rate, $V_\text{true}$ denotes the true velocity, and $V_\text{command}$ refers to the given velocity command. We greatly appreciate your constructive feedback, and to clarify our experimental setup, we will update the Appendix in the final version of the paper to include more detailed information on the real-world experiments.
> > >
> > > > Why observation history is not used?
> > >
> > > We would like to provide further clarification on this point. The introduction of historical information is primarily aimed at addressing the issue of privileged information that cannot be directly accessed in the Actor network of POMDPs. By utilizing the historical observations of the agent, we can estimate the environment's privileged information, thereby tackling the sim-to-real problem. Previous work has already demonstrated the effectiveness of using historical information to improve sim-to-real performance, but this is not within the scope of the novelty of our paper. Our approach, through distributional reinforcement learning, explicitly models the risks posed by the surrounding environment (e.g., external disturbances, sensor noise, etc.) during the robot's walking process. This guides the robot to explore the action space in a safe and efficient manner during training, leading to better generalization and robustness. In the Appendix, we have experimentally shown that our method has a beneficial effect on motion control. Even without using historical information, our approach still achieves relatively good results. In short, historical information and our method address two different problems. Therefore, we believe that the use of historical information is not part of the novelty of this paper.
> > >
> > > We hope we have been able to answer your questions and provide a helpful perspective on this paper’s contribution. Please kindly let us know if you have additional questions or concerns that stand between us and a higher score!
> > >
> > >
> > > Once again, thank you for your time and effort in reviewing our work. We look forward to your feedback.

---

> > > > ### Author Response · Authors · 2024-12-04
> > > > **Response to Reviewer 18Yh**
> > > >
> > > > Dear Reviewer 18Yh,
> > > > ﻿
> > > >
> > > > Thank you very much for your insightful comments and suggestions. We have carefully considered each point and provided detailed responses as outlined above. We hope that our clarifications and improvements address your concerns and strengthen the manuscript.
> > > > ﻿
> > > >
> > > > We kindly request that you take the time to review our rebuttal and consider re-evaluating your initial score in light of the changes we have made. Your feedback is invaluable to us, and we are committed to producing the highest quality research.
> > > > ﻿
> > > >
> > > > Thank you again for your time and consideration.
> > > >
> > > > Kind regards,
> > > >
> > > > ﻿ Authors

---

> > > ### Author Response · Authors · 2024-12-03
> > > **Response to Reviewer 18Yh**
> > >
> > > Dear Reviewer 18Yh,
> > >
> > > ﻿ We sincerely appreciate your insightful comments and valuable suggestions on improving our work. As the final deadline approaches, we would kindly like to ask whether our responses have sufficiently addressed your questions and concerns. We are happy to engage in further discussion or provide any clarifications needed, and we welcome any additional feedback to strengthen our work before the rebuttal period ends.﻿ Please kindly let us know if you have additional questions or concerns that stand between us and a higher score! Thank you for your time and thoughtful reviews. ﻿
> > >
> > > Kind regards,
> > >
> > > ﻿ Authors

---

> > > ### Author Response · Authors · 2024-12-03
> > > **Response to Reviewer 18Yh**
> > >
> > > Dear Reviewer 18Yh,
> > >
> > > First and foremost, we sincerely thank you for your valuable feedback on our paper. Based on your suggestions, we address the concerns you raised：
> > >
> > > ● We have elaborated the details of the experimental settings in detail.
> > >
> > > ● We have explained why our method did not use historical information.
> > >
> > > Once again, thank you for your time and effort in reviewing our work. We look forward to your feedback. We hope we have been able to answer your questions and provide a helpful perspective on this paper’s contribution. Please kindly let us know if you have additional questions or concerns that stand between us and a higher score!
> > >
> > > Best regards,
> > >
> > > Authors of 14098

---

### Author Response · Authors · 2024-11-28
**Further Discussion**

Dear Reviewers,

We sincerely appreciate your thoughtful comments and valuable suggestions. As the final revision deadline is approaching, we would like to kindly ask if our responses have adequately addressed your concerns. We remain open to further discussion or clarification and welcome any additional feedback to improve our work before the submission deadline.

Thank you again for your time and insights.

Best regards, Authors of 14098

---

### Author Response · Authors · 2024-12-04
**Rebuttal Summary**

We sincerely thank all the reviewers for their valuable comments and for taking the time to review our manuscript. These constructive suggestions have greatly helped us improve the content of the paper. In the revised manuscript, we have provided detailed responses to all comments. Below is a summary of our responses to the reviewers' comments:

1. **Motivation and role of the RND module**: In the revised version, we clarified the motivation for using the RND module in **lines 300-303**. Additionally, we discussed the role of the RND module in the ablation experiments provided in **Appendix A.5**, further strengthening our claims.

2. **Validation of the effectiveness of the Wang_function**: We included additional experiments in **Appendix A.8**, testing the CVaR distortion function combined with the IQR and RND modules. The results show that CVaR consistently failed to train under various hyperparameter configurations, verifying that the Wang function is more compatible with our framework. Furthermore, we analyzed the reasons behind CVaR's failure in detail.

3. **Supplementing real-world experiments**: We conducted three additional quantitative experiments on real-world robots and provided detailed analyses in the revised manuscript. First, we employed a pendulum system to generate a fixed lateral impact force and tested the success rate of the robot at a velocity of 0.6 m/s. A failure was defined as the robot falling over. Experimental results demonstrated that our method achieved the highest success rate, outperforming other methods. Additionally, we evaluated the velocity error rate under the condition of adding a 5 kg load to the robot's center of mass. Through these experiments, we further assessed the impact of load on the robot's stability. Another setting of experiments was designed to apply tensile force on the robot's legs by adding loads to its feet, testing the velocity tracking error rate under this condition. The results showed that, in both scenarios, our method consistently maintained the lowest velocity error rate, demonstrating superior robustness and stability. These experiments presented in **Section 4.3** verified the effectiveness of our method under complex conditions. We firmly believe that these results clearly highlight the innovation and superiority of our approach, fully addressing the reviewers' concerns.

4. **Enhancing reliability**: In the revised version, we explicitly stated the number of random seeds and repetitions used for each experiment to enhance the robustness and reliability of our results. We also described the training and testing setups in detail in **Appendix A.3**, particularly focusing on evaluations in out-of-distribution scenarios.

5. **Independence on reward structures**: To demonstrate that our method does not depend on specific reward structures, we conducted training and testing on a quadruped robot using different reward formulations in **Appendix A.6**. The results indicate that, even with changes to the reward structure, our method consistently outperformed the baselines in terms of both rewards and success rate of survival.

6. **Improvements in clarity and readability**: We revised the organization of the text, corrected typos, optimized citation formatting, and ensured consistency in notation throughout the manuscript. We also improved the visuals (e.g., **Figure 2**) and included an algorithm block in **Appendix A.4** to explain the implementation of our method step by step.

We hope these improvements comprehensively address the reviewers' concerns. If there are any further questions or suggestions, please feel free to let us know. Thank you again for your invaluable feedback!

---

### Meta-Review · Area_Chair_oWim · 2024-12-22

**Metareview:**

This paper introduces HuRi, an adaptive risk-aware distributional reinforcement learning framework for humanoid robot locomotion. By combining intrinsic uncertainty (IQR) and parameter uncertainty (RND), the method dynamically adjusts risk sensitivity to improve robustness in uncertain environments. While the motivation and application to humanoid robots are relevant, the paper falls short in several key areas. First, the algorithmic novelty of the proposed framework is limited. The combination of IQR and RND, while interesting, primarily builds on existing techniques without offering significant new insights or advancements. Next, the claimed performance improvements, though measurable, are modest and do not convincingly justify the proposed approach as a major contribution to the field. The experimental validation, both in simulation and in the real world, is insufficient to substantiate the paper’s claims. The real-world experiments, while appreciated, lack statistical rigor, with limited repetitions and insufficient detail on key metrics. Moreover, there is lack of comparisons with baselines, and the evaluations fail to demonstrate the generalizability and broader applicability of the method. Without robust experimental evidence, the method's efficacy remains unclear.

Additionally, despite revisions during the rebuttal phase, the paper’s presentation still lacks clarity in key areas, particularly in describing the methodology and experimental setup. Ambiguities in mathematical notation and incomplete explanations of components like SR(λ) and the Wang function make the method difficult to evaluate or reproduce. Finally, the paper’s focus on humanoid locomotion makes it more suitable for a specialized robotics venue than ICLR. The broader relevance of the approach to other machine learning domains is not sufficiently demonstrated. Given these limitations, the paper does not meet the standards for acceptance at ICLR in its current form.

**Additional Comments On Reviewer Discussion:**

During the discussion phase, reviewers raised consistent concerns about the paper’s algorithmic novelty, experimental rigor, and clarity of presentation. These points persisted despite the authors' efforts to address them during the rebuttal period.

- Algorithmic Novelty: Reviewer 7TAc noted that the proposed integration of IQR and RND lacked originality, stating, “While the paper attempts to solve a real-world problem, the presentation of the technical details and the underlying novelty are not sufficient. The framework primarily combines existing components without significant algorithmic advancements.” Reviewer 1gwC similarly commented, “The performance gain is not significant enough to justify a whole novel approach.” These concerns about limited novelty remained unresolved, as the authors primarily provided further clarifications without introducing substantial new ideas.

- Experimental Validation: Multiple reviewers emphasized the insufficient experimental validation. Reviewer 18Yh pointed out, “The real robot experiments are weak. There are no comparisons with robust baselines or statistical metrics to validate the claims of robustness.” While the authors included additional experiments during the rebuttal, they did not fully address these issues, as the new results lacked sufficient detail and statistical rigor. Reviewer 7TAc also noted, “Real-world experiments lack statistical or analytical significance. Success rates and safety violations should be reported to demonstrate the benefits of the risk-aware framework.”

- Clarity and Presentation: Several reviewers found the presentation unclear, with Reviewer 7TAc stating, “Throughout Section 3, the method is described mainly in text, making it difficult to understand. Figures and an algorithm block would have helped immensely.” Reviewer 1gwC similarly highlighted the lack of clarity in the mathematical descriptions, remarking, “Mathematical elements such as SR(λ) and the Wang function are ambiguously presented, making evaluation challenging.” Despite the authors’ revisions, these issues persisted in the updated manuscript.

- Broader Relevance: The reviewers questioned the paper’s fit for ICLR, with Reviewer 7TAc suggesting, “The focus on humanoid-specific locomotion makes this more suitable for a robotics venue than ICLR.” While the authors defended their focus, the broader applicability of the method to other machine learning domains was not convincingly demonstrated.

The reviewers appreciated the authors’ efforts to address the feedback, but their concerns about the novelty, rigor, and presentation remained largely unresolved.

---

### Decision · Program_Chairs · 2025-01-22

Reject